# A monoclonal antibody targeting a large surface of the receptor binding motif shows pan-neutralizing SARS-CoV-2 activity

Leire de Campos-Mata[1,11,14], Benjamin Trinité[2,14], Andrea Modrego[3,14], Sonia Tejedor Vaquero[1], Edwards Pradenas[2], Anna Pons-Grífols[2], Natalia Rodrigo Melero[4], Diego Carlero[3], Silvia Marfil[2], César Santiago[3], Dàlia Raïch-Regué[2], María Teresa Bueno-Carrasco[3], Ferran Tarrés-Freixas[2,12], Ferran Abancó[2], Victor Urrea[2], Nuria Izquierdo-Useros[2,5], Eva Riveira-Muñoz[2], Ester Ballana[2,5,6], Mónica Pérez[7,8], Júlia Vergara-Alert[7,8], Joaquim Segalés[7,9], Carlo Carolis[4,15] ✉, Rocío Arranz[3,15] ✉, Julià Blanco[2,5,6,10,15] ✉ & Giuliana Magri[1,13,15] ✉

Here we report the characterization of 17T2, a SARS-CoV-2 pan-neutralizing human monoclonal antibody isolated from a COVID-19 convalescent individual infected during the first pandemic wave. 17T2 is a class 1 VH1-58/κ3-20 antibody, derived from a receptor binding domain (RBD)-specific IgA⁺ memory B cell, with a broad neutralizing activity against former and new SARS-CoV-2 variants, including XBB.1.16 and BA.2.86 Omicron subvariants. Consistently, 17T2 demonstrates in vivo prophylactic and therapeutic activity against Omicron BA.1.1 infection in K18-hACE2 mice. Cryo-electron microscopy reconstruction shows that 17T2 binds the BA.1 spike with the RBD in "up" position and blocks the receptor binding motif, as other structurally similar antibodies do, including S2E12. Yet, unlike S2E12, 17T2 retains its neutralizing activity against all variants tested, probably due to a larger RBD contact area. These results highlight the impact of small structural antibody changes on neutralizing performance and identify 17T2 as a potential candidate for future clinical interventions.

Severe acute respiratory syndrome coronavirus 2 (SARS-CoV-2), the etiological agent of Coronavirus disease 2019 (COVID-19), has provoked one of the worst pandemics in human history, causing more than 6.9 million deaths registered worldwide (https://covid19.who.int/). The high level of virus circulation amongst humans and other species has led to the emergence of several variants with progressively increased transmissibility and immune evasion capacity[1–3]. In December 2021, Omicron became the globally dominant circulating strain, after replacing previous variants. The initial Omicron wave was caused by the BA.1 variant, followed by several Omicron sublineages (BA.2, BA.4, BA.5 and BQ.1.1, among others). Compared with the ancestral strain identified in Wuhan

(WH1), the spike protein of the Omicron lineage contains at least 30 amino acid substitutions, which are largely confined to the receptor binding domain (RBD) and the N-terminal domain (NTD), the two major antigenic sites targeted by neutralizing antibody response[4]. More recently, the diversification of Omicron sublineages has led to the emergence of variants like XBB.1.5[5], XBB.1.16[6], and EG.5.1[7], which have sequentially become dominant in many countries. These variants accumulate further mutations, including the F486P substitution, and have rapidly become a global public health concern due to an increase in SARS-CoV-2 infections and reinfections in developed countries[8–10]. Furthermore, a highly divergent BA.2-derived variant called BA.2.86,

which also carries the F486P mutation, has been identified in August 2023. This variant shows significantly higher RBD-ACE2 binding affinity and transmissibility compared to previous subvariants and emerges as a new global threat[11,12]. The continuous evolution of SARS-CoV-2 has not only increased the transmissibility of these newer variants but has also caused a considerable resistance to vaccine-induced antibody responses, leading to a surge in vaccine breakthrough infections worldwide[3,13]. Together with vaccines and antiviral drugs, neutralizing monoclonal antibodies (mAbs) targeting the spike protein of SARS-CoV-2 have been extensively used to treat patients at the highest risk of severe COVID-19 and to protect immunocompromised individuals from infection[14,15]. During previous COVID-19 waves, administration of mAbs was reported to be highly effective in preventing COVID-19-related infections, hospitalization and death[16,17]. However, most of the mAbs that have been initially approved for emergency clinical use were developed against ancestral SARS-CoV-2 and they all lost or significantly reduced their activity against highly mutated Omicron sublineages[1,3,18]. Although one newly developed antibody is currently in clinical development[19] (clinical trial NCT05872958), there is still an urgent need to develop pan-SARS-CoV-2 neutralizing antibodies that are effective against current and future variants. These antibodies will provide additional therapeutic options to patients and will inform on the existence of highly conserved vaccinable motifs in the SARS-CoV-2 spike.

Here we report the functional and structural characterization of a human mAb developed from a patient infected with the ancestral SARS-CoV-2 variant that shows pan-neutralizing activity against both pre-Omicron and latest Omicron SARS-CoV-2 variants.

## Results

### Generation of human recombinant SARS-CoV-2-specific monoclonal antibodies

To generate human mAbs capable of neutralizing SARS-CoV-2, we sorted 380 circulating RBD+ B cells from a convalescent COVID-19 individual who was infected during the first wave of the pandemic in Spain[20], using a biotinylated RBD protein from the ancestral SARS-CoV-2 strain as bait (Supplementary Fig. 1a). RBD-specific B cells were further characterized according to the expression of CD27, CD21, CD11c, HLA-DR, IgM, IgD, IgA and the Ig light chain λ (Supplementary Fig. 1b). mRNA from sorted cells were then reverse-transcribed and the immunoglobulin heavy chain (IGHV) and light chain variable (IGLV) regions were amplified by PCR following an established protocol[21]. After Sanger sequencing, 5 RBD-specific IGHV and IGLV paired regions were cloned into expression vectors and generated as recombinant human monoclonal IgG1. Three of these mAbs were generated starting from RBD-specific CD21+CD27+ canonical IgG memory B cells whereas two of them, 17T2 and 54T1, were generated starting from RBD-specific IgA canonical memory (IgA+ ME) B cells (Supplementary Table 1). As expected, all 5 mAbs showed relatively low levels of somatic mutations, which is consistent with the low level of hypermutation reported in RBD-specific antibodies following infection with the ancestral SARS-CoV-2[22] (Supplementary Table 1). These mAbs were screened to confirm their reactivity profile by enzyme-linked immunosorbent assay (ELISA) using recombinant RBD from the WH1 and the subsequent Beta, Gamma, Delta, Omicron BA.1 and Omicron BA.2 variants as immobilized proteins. As expected, all mAbs bound to the RBD from the ancestral variant, yet only 17T2 was able to efficiently recognize all variants tested, including RBD from highly mutated Omicron variants (Supplementary Fig. 2). The binding affinity of 17T2 for the RBD from different SARS-CoV-2 variants was then assessed by surface plasmon resonance (SPR). This analysis confirmed the high-affinity binding with equilibrium dissociation constants ($K_D$) in the subnanomolar range for all variants tested (Supplementary Table 2). Interestingly, 17T2 belongs to the IGHV1-58/κ3-20 clonotype, as many other potent neutralizing mAbs isolated from SARS-CoV-2-infected and/or vaccinated individuals[23–28] (Supplementary Table 1).

### 17T2 mAb shows high neutralization activity against SARS-CoV-2 variants including Omicron sublineages

To evaluate the functional activity of the selected antibodies, we tested their neutralization capacity using HIV reporter pseudoviruses expressing different SARS-CoV-2 spike proteins from variants ranging from WH1 to Omicron BA.1 (Supplementary Table 3)[29]. Consistent with the reactivity data, all antibodies showed neutralization of WH1 and D614G pseudoviruses. However, only 17T2 mAb maintained high neutralizing capacity against all variants tested, while the other mAbs markedly lost potency against Beta, Gamma, Mu or Omicron BA.1 variants (Fig. 1a and b). 131T2 showed the lowest potency against pre-Omicron variants and no activity against Omicron BA.1 (Fig. 1b). To further characterize 17T2 mAb, we analyzed neutralization of pseudoviruses exposing the spike of newer Omicron subvariants (Supplementary Table 3) and SARS-CoV-1. Moreover, for comparative purposes, we assayed in parallel two well-characterized broad-spectrum neutralizing RBD-targeting mAbs: S2E12 and S309[28,30]. S2E12 is a VH1-58/κ3-20-encoded class 1 antibody, like 17T2, whose antibody-binding epitope overlaps with the receptor binding motif (RBM) in the RBD[27], whereas S309 is a class 3 antibody isolated from a patient recovering from SARS, which binds to a conserved epitope outside the RBM[31]. 17T2 mAb neutralized all SARS-CoV-2 variants tested (pre-Omicron variants WH1, D614G, Alpha, Beta, Gamma, Delta, and Omicron subvariants BA.1, BA.2, BA.4/5, BQ.1.1, XBB.1.5, XBB.1.16, EG.5.1, and BA.2.86). $IC_{50}$ values were in the low ng/mL range for most pre-Omicron and early Omicron variants, with higher values for the latest Omicron XBB.1.5, XBB.1.16, BA.2.86 subvariants ($IC_{50}$s ranging from 387 to 541 ng/mL), and the highly mutated EG.5.1 ($IC_{50}$ 1180 ng/mL). In contrast, no activity against SARS-CoV-1 was observed (Fig. 1c and d). The wide SARS-CoV-2 neutralization spectrum of 17T2 was remarkably different from the other two broadly neutralizing mAbs. S2E12 showed higher potency against all pre-Omicron variants but had significant lower neutralizing activity against BA.1, BA.2 and no activity against BA.4/5 and all subsequent variants (Fig. 1c and d), while S309 antibody maintained some activity against Omicron subvariants except for BA.2.86 (Fig. 1c-e).

Comparable results were obtained when the 17T2, S309, and S2E12 mAbs were tested against a large panel of pre-Omicron and Omicron SARS-CoV-2 primary isolates (from D614G to BQ.1.1; Supplementary Fig. 3), confirming the broader neutralization capacity of 17T2 compared to the structurally related antibody S2E12.

### 17T2 mAb shows prophylactic and therapeutic activity against Omicron BA.1.1 in vivo in K18-hACE2 transgenic mice

Next, we tested 17T2 in vivo prophylactic efficacy in K18-hACE2 transgenic mice. 17T2 mAb or an isotype control antibody (IgGb12) were administered intraperitoneally (10 mg/kg) 24 h before challenge with $10^3$ 50% Tissue Culture Infectious Dose (TCID50) of a BA1.1 SARS-CoV-2 isolate (Fig. 2a). As previously reported[32,33], Omicron infection of K18-hACE2 transgenic mice resulted in a mild disease without significant changes in weight in infected animals as compared to uninfected ones (Fig. 2b). However, isotype control-treated animals showed high viral loads in oropharyngeal swabs, lungs and nasal turbinates, both 3- and 7-days post-infection (dpi) (Fig. 2c). In comparison, mice treated with 17T2 mAb showed significantly lower viral loads in lungs 3 dpi and in all tissues assayed (oropharyngeal swabs, lungs, and nasal turbinate) 7 dpi (Fig. 2c). The protective effect of 17T2 was confirmed in lung tissue by analyzing the histopathological lesions and the SARS-CoV-2 nucleocapsid focal expression at 3 and 7 dpi by histology and immunohistochemistry, respectively (Fig. 2d, e).

To demonstrate the therapeutic activity of 17T2 mAb we used a similar experimental model of infection: K18-hACE2 mice were challenged with a BA.1.1 isolate and 24 h later, either the 17T2 mAb or an isotype control antibody (IgGb12) were administered intraperitoneally

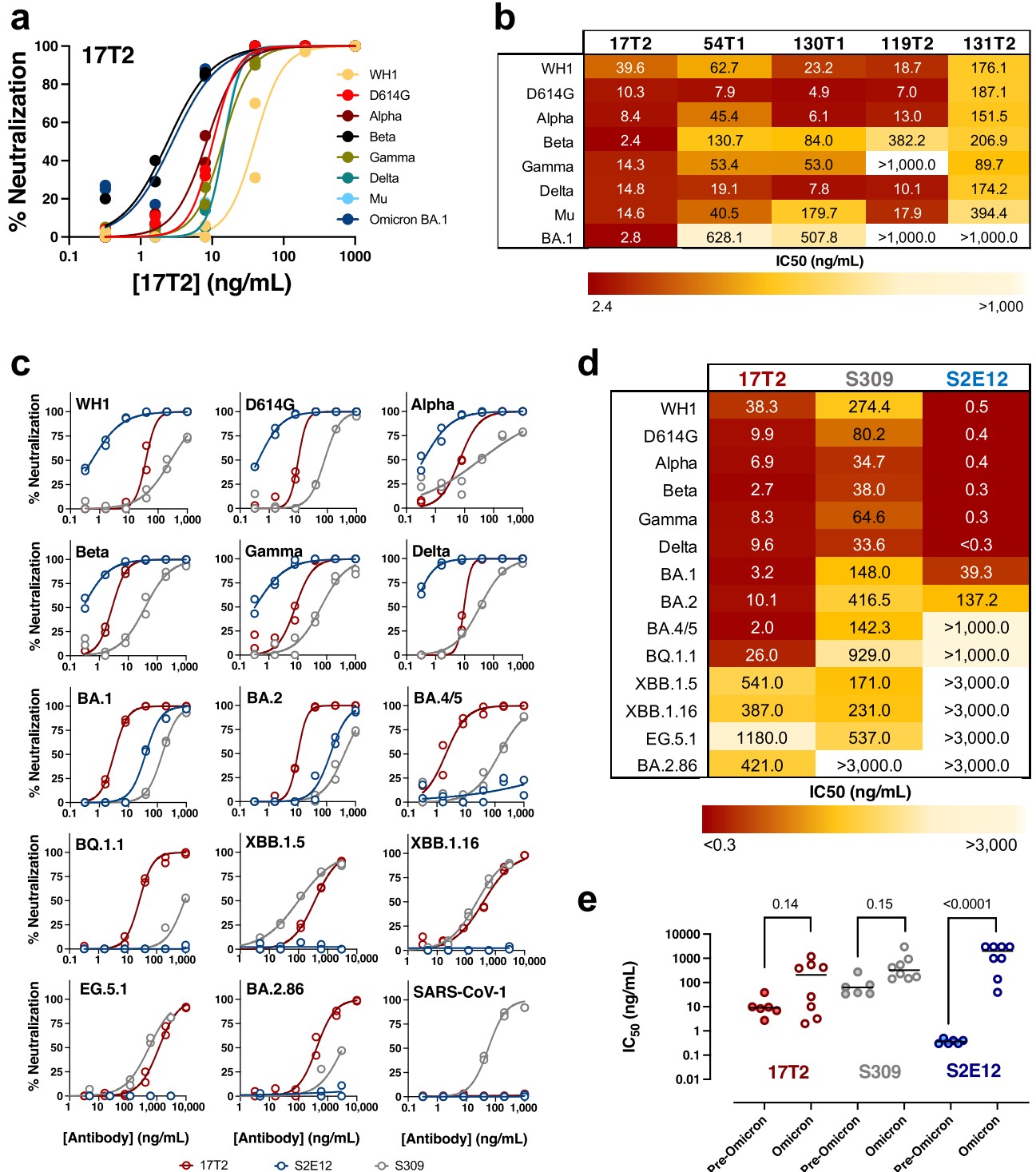

**Fig. 1 | Pan-neutralizing activity of 17T2 mAb. a** Neutralization curves of 17T2 mAb against HIV-1 based pseudoviruses expressing the indicated SARS-CoV-2 spike variants. Duplicate values corresponding to a representative experiment out of two are shown. **b** Heatmap showing the neutralization activity as $IC_{50}$ values of the selected mAbs against the indicated SARS-CoV-2 pseudoviruses. Values are in ng/mL, darker color corresponds to higher potency as indicated in the bottom of the Figure. **c** Neutralization curves of 17T2 (red), S309 (gray) and S2E12 (blue) mAbs against the indicated SARS-CoV-2 variants or SARS-CoV-1 (all exposed on the surface of HIV-1 based pseudoviruses). Duplicate values corresponding to a representative experiment out of at least two are shown. **d** Heatmap showing $IC_{50}$ values from panel C in ng/mL. As in (**b**), darker color corresponds to higher potency as indicated in the bottom of the Figure. **e** Impact of Omicron subvariants on pseudovirus neutralization capacity. $IC_{50}$ values for pre-Omicron variants (WH1, D614G, Alpha, Beta, Gamma, and Delta) were grouped ($n = 6$) and compared to Omicron subvariants (BA.1, BA.2, BA.4/5, BQ.1.1, XBB.1.5, XBB.1.16, EG.5.1 and BA.2.86; $n = 8$). Solid bars show the geometric mean. *P* values show individual corrected comparisons for each antibody using two-sided Kruskal-Wallis test with a global *p* value of <0.0001. Source data are provided as a Source Data file.

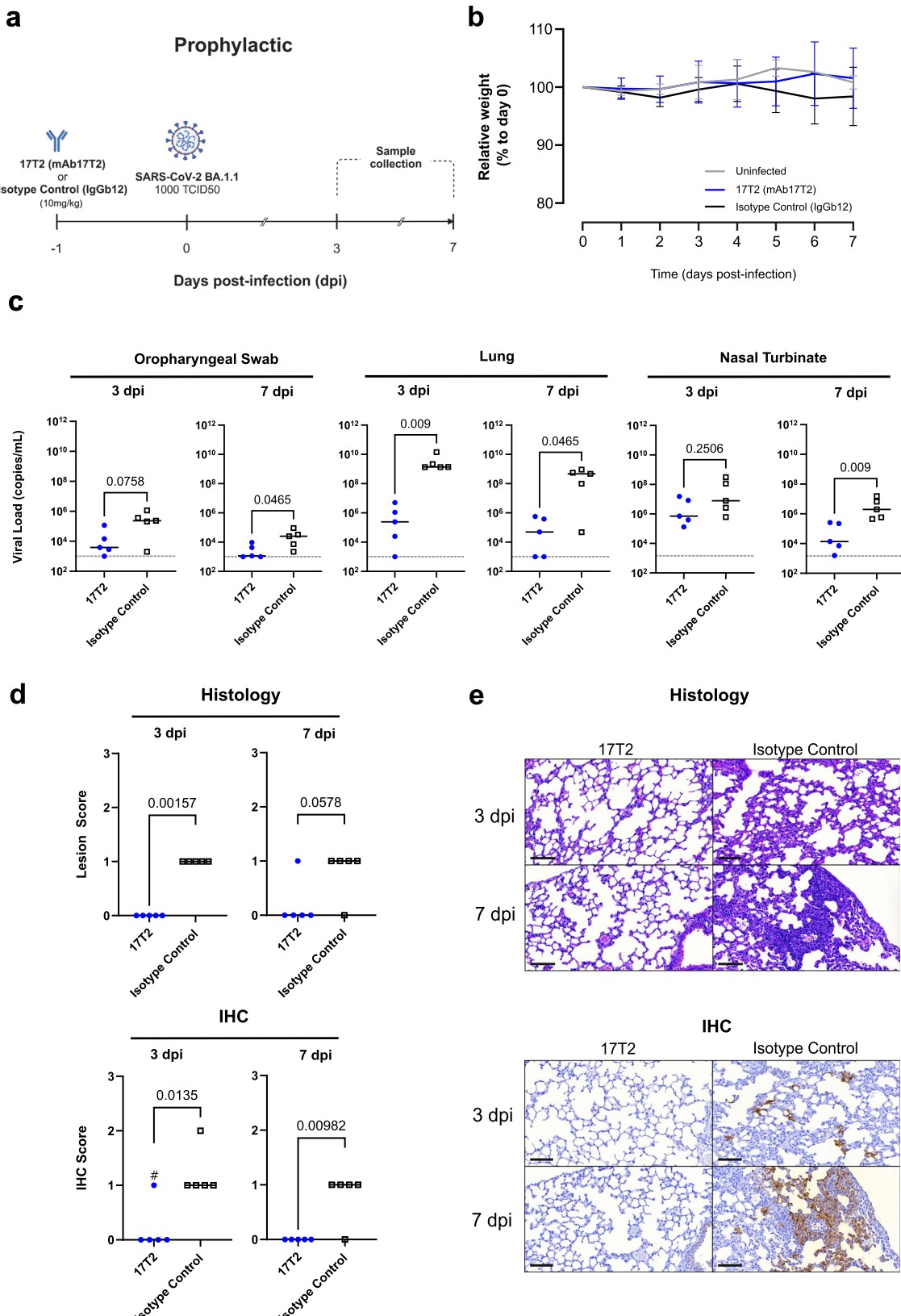

(30 mg/kg) (Fig. 3a). Again, the animals did not show any relevant weight changes (Fig. 3b). Nevertheless, 17T2 treatment significantly reduced the viral loads in oropharyngeal swabs at 5 dpi compared to the isotype control antibody ($p = 0.0056$), while it displayed a more modest effect in lungs ($p = 0.0830$, Fig. 3c). The effect in the lungs was more evident when we assessed viral infectivity, since 17T2 efficiently

reduced lung viral titers compared to the isotype control ($p = 0.0032$, Fig. 3d). Consistent with virological data, immunohistochemistry analysis of lungs revealed the presence of residual viral antigens in most treated animals (Fig. 3e, f). However, the histological analysis s showed a significant protective effect of 17T2 on tissue lesions, which were absent in most 17T2-treated animals while present in most

**Fig. 2 | Prophylactic 17T2 protection against SARS-CoV-2 BA1.1 infection in K18-hACE2 transgenic mice. a** Schematic description of the prophylactic experimental setting. Transgenic K18-hACE2 mice were administered intraperitoneally with 10 mg/kg of 17T2 (17T2 mAb, $n = 10$) or an isotype control (IgGb12, $n = 10$). After 24 h, treated animals were intranasally challenged with an Omicron BA.1.1 SARS-CoV-2 isolate ($n = 20$), or PBS (Uninfected Control Group) ($n = 4$). Mice were monitored for 7 days. Euthanasia was performed 3- and 7-days post-infection (dpi) ($n = 5$ for each treated group per timepoint, $n = 2$ uninfected per timepoint) for sample and tissue collection. Created with Biorender.com. **b** Relative K18-hACE2 transgenic mice weight-loss follow-up. Groups include uninfected (gray, $n = 4$), 17T2 mAb-treated and infected (blue, $n = 10$), and isotype control-treated and infected (black, $n = 10$) animals. Solid lines represent mean ± SD. **c** SARS-CoV-2 viral RNA load quantification (copies/mL) on oropharyngeal swab and lung at 3 and 7 dpi in 17T2 mAb-treated (blue dots, $n = 5$ per timepoint) or isotype control-treated (black squares, $n = 5$ per timepoint) infected animals. The limit of detection is represented by a dashed line. Statistical differences were determined using a two-sided Peto-Prentice generalized Wilcoxon test. **d** Histopathological and immuno-histochemical scores of lungs from infected and prophylactically treated K18-hACE2 mice. Lesion (broncho-interstitial pneumonia) scoring: 0 = no lesion, 1 = mild lesion, 2 = moderate lesion, 3 = severe lesion. IHC scoring: 0 = no antigen, 1 = low and multifocal antigen, 2 = moderate and multifocal antigen, 3 = high and diffuse antigen. # indicates an animal which presented focal expression only visible on 1 out of 5 lung sections: it was scored as 1 but with minimal detection of virus replication. All other positive scores showed multifocal distribution on multiple lung sections. Comparisons were performed using a two-sided Asymptotic Generalized Pearson Chi-Squared Test for ordinal data with pairwise comparisons. **e** Representative histopathological and immunohistochemical findings, at 3 and 7 dpi, in K18-hACE2 transgenic infected mice treated with either 17T2 mAb or an isotype control. Histological slides were stained with hematoxylin and eosin, and immunohistochemistry ones were counterstained with Hematoxylin. Scale Bar = 80 μM. Source data are provided as a Source Data file.

control animals ($p = 0.010$, Fig. 3e and f), confirming the therapeutic efficacy of the antibody.

## 17T2 binds the Omicron BA.1 spike protein with the RBD domains in the up position and recognizes a large surface overlapping with the receptor binding motif

We carried out cryogenic electron microscopy (cryo-EM) analysis to understand the potency and breadth of 17T2-mediated neutralization of SARS-CoV-2 variants. For this reason, we solved the structure of the complex between the highly mutated Omicron BA.1 trimeric spike and the 17T2 Fab fragment, reaching a resolution of 3.46 Å (Fig. 4a and b). Our analysis showed that 17T2 Fab binds to RBD in the "up" conformation with all particles containing 3 17T2 Fabs, each one bound to adjacent RBDs within a single spike trimer (Fig. 4a and b). Due to the higher conformational dynamics in the 17T2 variable domains and the RBD regions, resolution in the contact area was lower than in the rest of the trimer. Local refinement was performed in this region, significantly improving local resolution to 4.41 Å (Fig. 4c and Supplementary Fig. 4). After refinement, we determined that 17T2 Fab binds to the left shoulder-neck region of the RBD that is only accessible in "up" conformation (Fig. 4b–d). The interaction site overlaps with the RBD in a similar manner to other class 1 VH1-58/κ3-20-derived neutralizing mAbs[23–28]. Fab 17T2/RBD interactions involve both the heavy chain (HC) and light chain (LC) of the antibody, covering 563Å² for the HC and 295 Å² for the LC of the total interaction surface (Fig. 4d). 17T2 Fab uses complementary determinant regions (CDR) 1 to 3 of the HC and CDR1 and CDR3 of the LC to recognize residues 420 to 421; 455; 473 to 478; 480; 484 to 487, 489 and 493 of the SARS-CoV-2 RBD (Fig. 4d, e). The contact area mostly involved Van der Waals interactions with a minor contribution from hydrogen bridges (Fig. 4e). In addition, we identified a salt bridge formed between the D420 residue of the RBD and the R103 residue of H3 that stabilizes the binding of 17T2 to the RBD (Fig. 4e). As previously described for structurally similar neutralizing antibodies[24,34], 17T2 is glycosylated at the N102 residue of H3 located near the left shoulder of the RBD (Fig. 4c). Using Peptide:N-glycosidase-mediated de-glycosylation of 17T2 mAb (17T2 De-gly) (Supplementary Fig. 5a) we confirmed that this glycosylation had no effect on the affinity nor the neutralizing activity of 17T2 mAb (Supplementary Fig. 5b–d).

The 17T2 Fab binds to a large and mostly conserved area of RBD, with mutated residues harbored by SARS-CoV-2 variants located at the edge of the contact surface (Fig. 4f and Supplementary Table 4). Since 17T2 and S2E12 mAbs share high sequence identity but differ in their neutralizing activity against highly mutated Omicron subvariants, we compared the structures of their respective Fabs/spike complexes (Supplementary Fig. 6a and b). This comparison revealed that S2E12[24,27,34] and 17T2 Fabs bind parallel to the longest axis of the hACE2 binding site. Nevertheless, the area of interaction between 17T2 and RBD is broader than the area between S2E12 and RBD, the latter being included in the 17T2 interaction area (Supplementary Fig. 6c). Moreover, although the structure of the two antibodies is highly similar, the amino acid side chains of 17T2 are located closer to the surface of the RBD (Supplementary Fig. 6b), allowing for higher number of contacts with conserved residues. This fact probably contributes to the ability of this antibody to neutralize Omicron subvariants exposing S477N and F486V mutations, which strongly impact S2E12 binding.

## Discussion

Here we describe the functional and structural characterization of 17T2, a human mAb with broad neutralizing activity against all SARS-CoV-2 variants tested, including early Omicron sublineages BA.1, BA.2, BA.4, BA.5, BQ.1.1, and the later subvariants XBB.1.5, XBB.1.16, and BA.2.86. Importantly, both prophylactic and therapeutic administration of 17T2 mAb resulted in a significant reduction of microscopic lung lesions in a mouse model of SARS-CoV-2 Omicron infection.

17T2 belongs to the class 1 VH1-58/κ3-20-derived antibodies, which include several mAbs (i.e., S2E12, Cv2.1169, A23.58.1, AZD8895) with high neutralizing activity against pre-Omicron variants[24–28]. However, compared to these structurally similar mAbs, 17T2 retains its potency more effectively against Omicron sublineages, including the increasingly immune evasive variants BA.5, BQ.1.1, XBB.1.5, XBB.1.16, and BA.2.86, with IC50 values below 0.6 μg/mL. The lowest activity of 17T2 mAb was observed against EG.5.1 (1.2 μg/mL). The structural analysis of 17T2 Fab in complex with Omicron BA.1 spike trimer suggests complementary mechanisms to explain its broad neutralizing activity. On the one hand, the high antibody affinity allows for a complete blockade of all RBDs of the spike trimer stabilized in the "up" conformation (stoichiometry 3 Fab:1 spike trimer). On the other hand, when compared to S2E12, 17T2 shows a larger area of interaction with the RBM, which could confer higher tolerability to RBD mutations. Moreover, the presence of a salt bridge between the R103 in the CDR H3 and the D420 in a conserved region of the RBD participates in the stabilization of the complex, contributing to the extraordinarily high affinity of 17T2 to the RBD from multiple SARS-CoV-2 variants. Interestingly, D420 has been recently identified by mutagenesis as a potential site for escaping neutralization by some class 1 antibodies[35,36]. The stability provided by D420 could explain why 17T2 mAb resists the F486V spike mutation present in the BA.4 and BA.5 variants which otherwise escape all other IGHV1-58-derived antibodies described thus far[3,26,37].

Comparative analyses between S2E12 and 17T2 indicate that single-point mutations in the CDRs of both the heavy and light chains of 17T2 confer unique structural features to the antibody, thus enhancing the potency and breadth of 17T2 in comparison to S2E12. However, the neutralization potency of 17T2 mAb varies across the tested variants (IC50 range: 3–1180 ng/μL). This variability can be

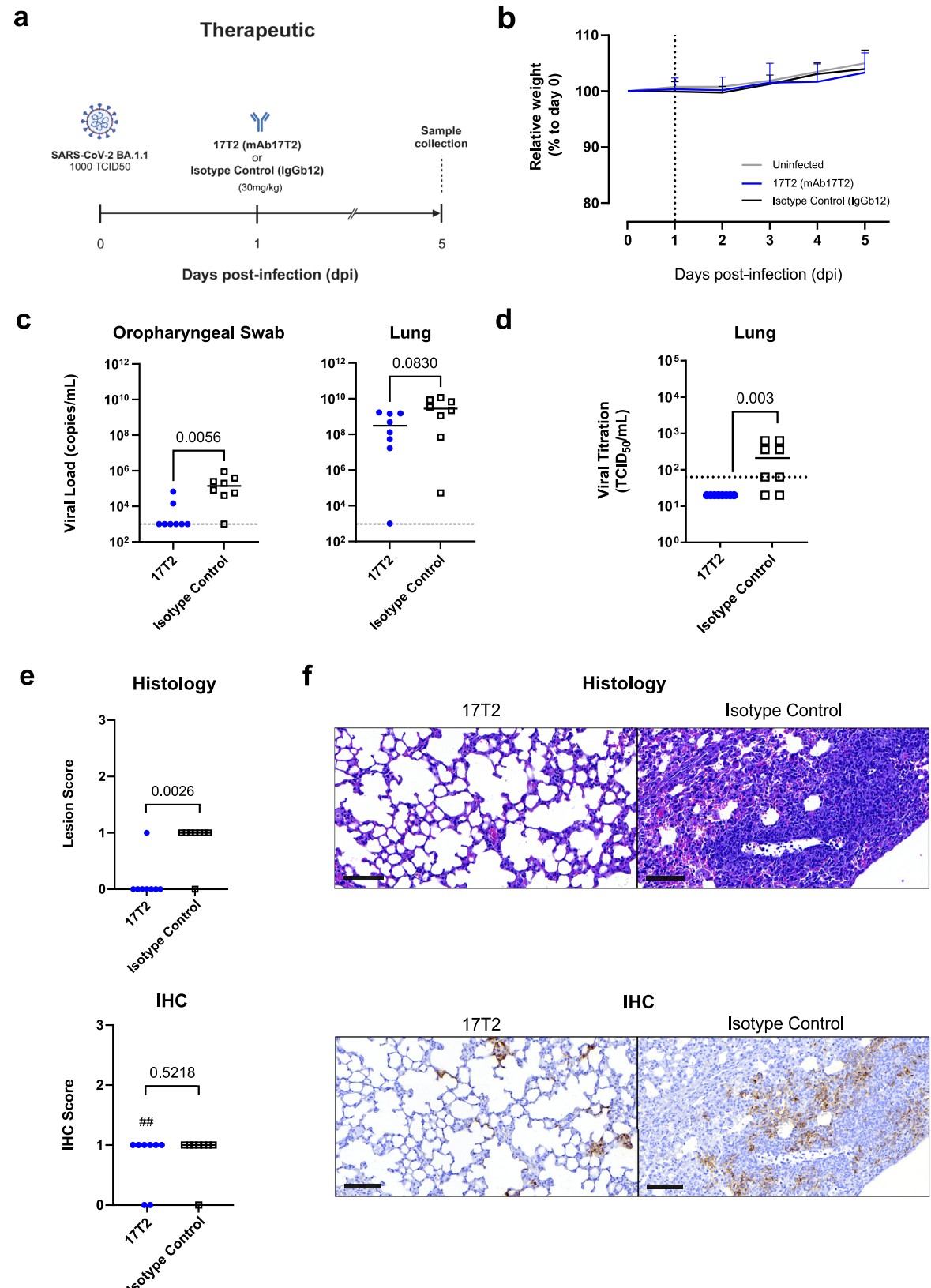

attributed to mutations influencing the RBD exposure in the spike trimer and the antibody binding area. For instance, a four-fold change in $IC_{50}$ value was observed when comparing WH1 and D614G pseudo-viruses, which share identical RBD sequences but differ in the RBD "up" and "down" conformational equilibrium. On the other hand, mutations present in the outer 17T2 mAb binding area, such as S477N, T478K,

E484A (common in most Omicron subvariants) and F486V (present in BA.4/5 and BQ.1.1) are well tolerated, suggesting some level of plasti-city in the mode of antibody binding. In contrast, the N460K and F486P mutations could be responsible for the reduced 17T2 neu-tralization capacity against the latest Omicron variants XBB.1.5, XBB.1.16, EG.5.1, and BA.2.86. N460K mutation may interact with the

**Fig. 3 | Therapeutic 17T2 protection against SARS-CoV-2 BA1.1 infection in K18-hACE2 transgenic mice. a** Schematic description of the therapeutic experimental setting. Transgenic K18-hACE2 mice were intranasally challenged with an Omicron BA.1.1 SARS-CoV-2 isolate ($n = 18$), or PBS (Uninfected Control Group) ($n = 2$). After 24 h, animals were administered intraperitoneally with 30 mg/kg of either 17T2 (17T2 mAb, $n = 8$) or an isotype control (IgGb12, $n = 8$). Mice were monitored daily for 5 days. Euthanasia of all animals was performed at 5 dpi for sample and tissue collection. Created with Biorender.com. **b** Relative K18-hACE2 transgenic mice weight-loss follow-up. Groups include: uninfected (gray, $n = 2$), 17T2 treated and infected (blue, $n = 8$), and Isotype Control treated and infected (black, $n = 8$) animals. Solid lines represent means. Error bars indicate ±standard deviation for 17T2 and Isotype control treated groups. **c** SARS-CoV-2 viral RNA load quantification (copies/mL) on oropharyngeal swab and lung 5 dpi in 17T2 mAb-treated (blue dots, $n = 8$ per timepoint), and isotype control-treated (black squares, $n = 8$ per timepoint) infected animals. Dashed line represents the limit of detection. Statistical differences were determined using a two-sided Peto-Prentice generalized Wilcoxon test. **d** Viral titration of replicative virus (TCID50) recovered from lung samples from 17T2 (17T2 mAb, $n = 8$) or an isotype control (IgGb12, $n = 8$) SARS-CoV-2-

infected mice at endpoint in Vero E6 cells at day 5 of culture. Comparison was determined using a two-sided Peto-Peto Left-Censored Two-Sample test. **e** Histopathological and immunohistochemical scores of lungs from infected and therapeutically treated K18-hACE2 mice. Lesion (broncho-interstitial pneumonia) scoring: 0 = no lesion, 1 = mild lesion, 2 = moderate lesion, 3 = severe lesion. IHC scoring: 0 = no antigen, 1 = low and multifocal antigen, 2 = moderate and multifocal antigen, 3 = high and diffuse antigen. ## indicates that six out of eight 17T2-treated animals which scored 1 for IHC showed lower positive signal compared to isotype control-treated animals that also scored 1 (low and multifocal antigen). Comparisons were performed using a two-sided Asymptotic Generalized Pearson Chi-Squared Test for ordinal data with pairwise comparisons. **f** Representative histo-pathological and immunohistochemical findings at 5dpi, in K18-hACE2 transgenic infected mice treated with either 17T2 mAb or an isotype control. Histological slides were stained with hematoxylin and eosin, and immunohistochemistry ones were counterstained with Hematoxylin. Pictures depict the mean scores for each technique (0 for 17T2 Histology and 1 for the rest of conditions and techniques). Scale Bar: 80 µM. Source data are provided as a Source Data file.

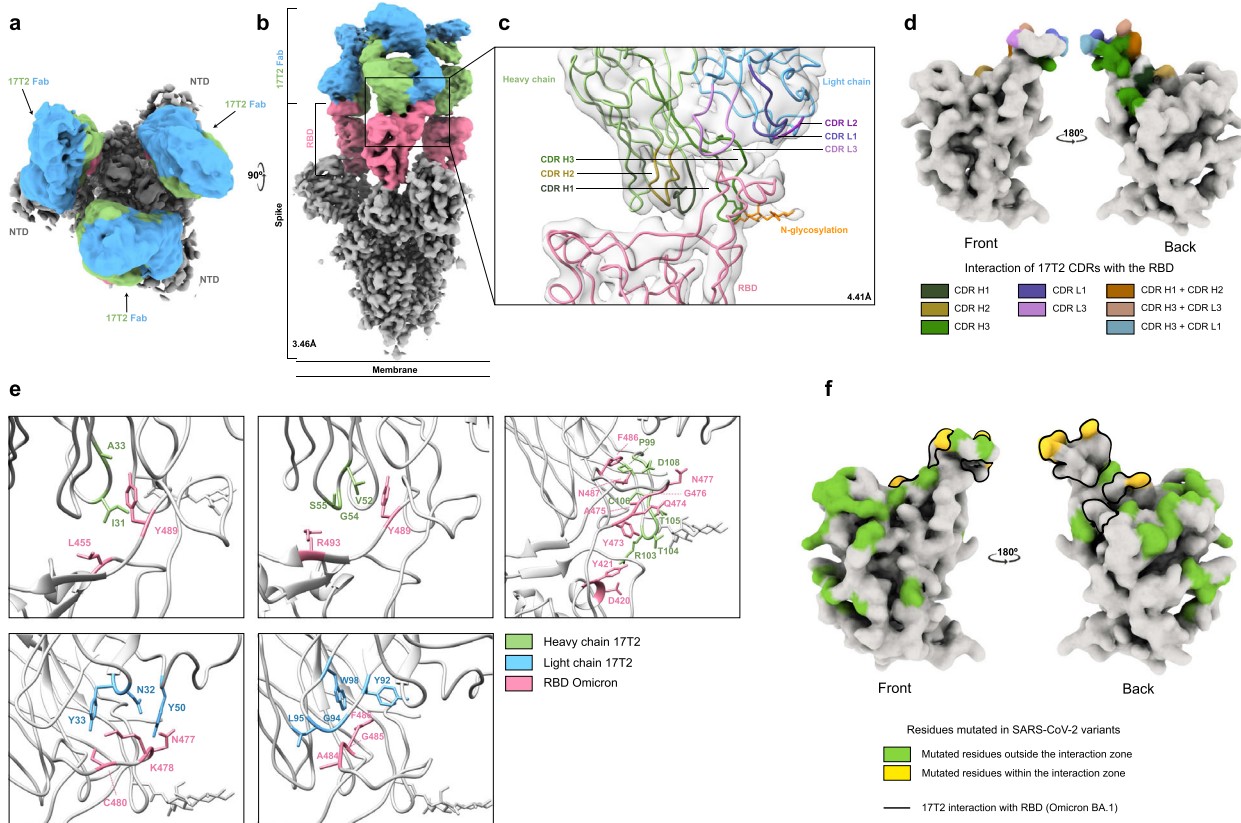

**Fig. 4 | Structural and functional characterization of complex 17T2 Fab fragment with Omicron BA.1 spike using cryo-EM. a, b** Top and side views of the cryo-EM map of Omicron BA.1 spike trimer with three 17T2 Fab fragments bound to three open RBDs, at 3.46 Å resolution. The core of the spike is shown in gray, the RBDs in pink, and the heavy chain (HC) and light chain (LC) of the 17T2 Fab in green and blue, respectively. **c** Structure of the RBD and 17T2 Fab after local refinement, at 4.41 Å resolution. The interaction zone between the RBD and the Fab is shown in cartoon representation where the three CDRs from each chain are distinctively colored and the N-glycosylation is indicated in orange. **d** CDRs that are involved in

binding with RBD, specifically, interacting in the region of its left shoulder and neck (front and back view, respectively). **e** A detailed view of some of the residues involved in the interaction between 17T2 Fab and RBD. The main chains are colored in gray and the side chains of the residues involved in the interaction are shown in green for HC, in blue for LC, and in pink for RBD. **f** Locations of SARS-CoV-2 variant mutations on RBD relative to 17T2 epitope site that is shown as a black line (front and back view, respectively). The information about the variants and mutated residues can be found in Supplementary Table 4.

residue 420D, as reported for the BA.2.75 variant[38], and impact the salt bridge formed between 420D and the R103 residue of the CDR H3 of the antibody. In addition, the F486P mutation, located on the opposite area of the antibody-RBD contact surface, could induce a higher constrain in the epitope flexibility compared to the previous F486V mutation, thereby modifying the overall neutralization capacity of

mAb. Finally, EG.5.1 carries one additional RBD mutation, F456L, which has been linked to reduced neutralization by certain type I antibodies[39]. In the case of 17T2, we observed that this mutation was associated with an increased $IC_{50}$, albeit not a complete loss of activity.

IgG Fab glycosylation is a key parameter in immunity with possible consequences on antigen binding and antibody activity[40]. Our

structural analysis revealed that 17T2 is glycosylated at the N102 residue in the CDR H3 region, in close proximity to the area of contact with the RBD. However, PNGase mediated de-glycosylation of 17T2 mAb had no effect on the binding to RBD or on the neutralizing activity, excluding a potential role of Fab glycosylation in its functional activity. Yet, we cannot rule out its possible implication in the stability or immune modulatory properties of the antibody[40].

All things considered, broadly active neutralizing mAbs with proven in vivo neutralizing activity, such as 17T2 mAb, are urgently needed to protect immunocompromised patients and those individuals at high risk of developing severe COVID-19. Only a few human mAbs have shown resilience against Omicron sublineages[41–43], among them antibodies SA58 and SA55 have shown partial or full coverage in the Omicron landscape[44], including the BA.2.86 subvariant[11]. Interestingly, these antibodies bind to the RBD in a region distant from the 17T2 epitope, and structural data[10] suggests that the RBD could simultaneously accommodate all three broadly neutralizing antibodies. Therefore, the combined use of 17T2 mAb with SA58 and SA55 could improve their clinical efficacy, as previously suggested for combinations of mAbs active against pre-Omicron variants[45].

The identification of an anti-RBM broadly neutralizing antibody carries significant implications for future COVID-19 pandemic management. First, 17T2 has been cloned from a circulating IgA⁺ memory B cell isolated from a convalescent COVID-19 patient infected with the ancestral SARS-CoV-2. Therefore, our study provides evidence that infection with the ancestral virus could elicit broadly neutralizing antibodies, likely of mucosal origin, against SARS-CoV-2 variants not yet circulating. Additionally, and considering the potent broad neutralizing activity of 17T2 mAb in vitro and in vivo, 17T2 mAb not only represents a promising candidate for future interventions, but also could inform on relevant conserved Spike epitopes that could be suitable targets for vaccination.

## Methods

### Ethical statement
All procedures involving human samples were approved by the Ethical Committee for Clinical Investigation of the Institut Hospital del Mar d'Investigacions Mèdiques (Number 2020/9189/I). Informed consent was obtained from the participants involved in the study.

All animal procedures were performed under the approval of the Committee on the Ethics of Animal Experimentation of the IGTP and the authorization of Generalitat de Catalunya (code: 11222).

### Human participants and sample collection
Blood samples were collected from a COVID-19 convalescent individual infected with SARS-CoV-2 during the first wave of the COVID-19 pandemic in Spain as previously described[20]. Diagnosis of SARS-CoV-2 infection was confirmed by reverse transcription–quantitative polymerase chain reaction (RT-qPCR) of nasopharyngeal swab. Peripheral blood mononuclear cells (PBMCs) were isolated from whole blood collected with EDTA anticoagulant via Ficoll–Paque Premium (GE Healthcare; cat. number: 17-5442-03) following the manufacturer's instructions. PBMCs were resuspended in fetal bovine serum (FBS; Gibco; cat. number: 16000-044) with 10% dimethyl sulphoxide (DMSO; Sigma-Aldrich) and stored in liquid nitrogen prior to use.

### Single B-cell FACS sorting
For the isolation of WH1 SARS-CoV-2 spike-specific B cells, 26.4 pmol His-Tagged Biotinylated SARS-CoV-2 (2019-nCoV) spike RBD Recombinant Protein (Sino Biological Inc.; cat. number: 40592-VO8H-B) was incubated for 1 h with 3.78 pmol Streptavidin Alexa Fluor 647 (Thermo Fisher Scientific; cat. number: S32357) and Streptavidin Alexa Fluor 488 (Thermo Fisher Scientific; cat. number: S32354), separately. Next, PBMCs were incubated with the fluorescently labeled RBD probes and with a cocktail of fluorescent conjugated antibodies containing anti-

CD19 Pe-Cy7 (Biolegend; cat. number: 302216, clone HIB19), anti-IgM BV605 (Biolegend; cat. number: 314524, clone MHM-88), anti HLA-DR AF700 (Biolegend; cat. number: 307626, clone L243), anti-CD38 APC-cy7 (Biolegend; cat. number: 303534, clone HIT2), anti Igλ light chain PerCP-Cy5.5 (Biolegend; cat. number: 316617, clone MHL-38), anti-IgD PE CF594 (BD Biosciences; cat. number: 562540, clone IA6-2), anti-CD21 PE-cy5 (BD Biosciences; cat. number: 551064, clone B-ly4) and anti-IgA Viogreen (Miltenyi; cat. number: 130-114-007, clone IS11-8E10). Dead cells were excluded through the use of 4'−6'-diamidine-2'-phenylindole (DAPI) (Sigma; cat. number: D9542). Alive DAPI⁻ CD19⁺RBD⁺ B cells were single-cell index sorted using a FACSAria II (BD Biosciences) into empty 96-well PCR plates (VWR). FACSDiva software (Becton Dickinson) was used for acquisition and FlowJo for post-sorting analysis. Plates were immediately sealed with foil, frozen on dry ice and stored at −80 °C.

### Expression-cloning of antibodies
Antibodies were identified and sequenced as previously described[21]. In brief, RNA from single cells was reverse transcribed in the original 96-well sort plate using random hexamers (Thermo Fisher Scientific), 0.76% NP 40 detergent solution (Thermo Fisher Scientific), RNasin ribonuclease inhibitor (Promega), DTT (Invitrogen) and Superscript III Reverse Transcriptase (Invitrogen, 180080-044). The resulting cDNA was stored at −20 °C for subsequent amplification of the variable IGH, IGL and IGK sequences by nested PCR and Sanger sequencing using the same reverse and forward primers as in[21]. Sequences were analyzed using Change-O (IgBlast). Following analysis, Ig sequence-specific PCR was performed for the successfully annotated transcripts. Amplicon from the first PCR reaction were used as templates for cloning into antibody expression vectors (Abvec2.0-IGHG1 for the heavy chain and Abvec1.1-IGKC or Abvec1.1.-IGLC2-XhoI for of the light chains, all from AddGene). Recombinant antibodies were produced by transient co-transfection into Expi293F human cells (Thermo Fisher Scientific) with purified DNA and polyethylenimine (PEI 1:3). Transfected cells were cultured for 5 days at 37 °C, 5% $CO_2$ and 80% humidity. Then cells were harvested and after centrifugation, the cell supernatants containing the secreted mAbs were purified by affinity chromatography in a HiTrap MabSelect (Cytiva) equilibrated in PBS and eluted with 10 mM Glycine at pH 3.

### Production of Fab
The purified 17T2 mAb was digested by incubation at 37 °C within the immobilized papain agarose resin according to the manufacturer's instructions (Thermo Fisher Scientific; cat. number: 20341). The fragment antigen-binding (Fab) part was separated from undigested IgG and the crystallizable fragment (Fc) using an Immobilized Protein A column (Thermo Fisher Scientific; cat. number: 20356). The flow through containing the Fab was concentrated through 10 kDa Amicon centrifugal filter units (EMD Millipore) and buffer exchange was performed with a 7 kDa molecular weight cut-off size exclusion resin (Thermo Fisher Scientific) to cryo-EM buffer (10 mM Tris at pH 7.6 and 20 mM NaCl).

### Production of recombinant SARS-CoV-2 proteins
The pCAGGS RBD construct, encoding for the RBD of the WH1 SARS-CoV-2 spike protein from the earliest lineage A virus (WH1, YP_009724390.1, residues 319-541; NC_045512.2, A lineage), along with the signal peptide plus a hexahistidine tag, was provided by Dr. Krammer (Mount Sinai School of Medicine, NY USA). RBD sequences from current Alpha, Beta, and Gamma variants were obtained from the World Health Organization tracking of variants (https://www.who.int/en/activities/tracking-SARS-CoV-2-variants/) and Pango lineage classification (https://cov-lineages.org/). DNA fragments encoding the RBD from Alpha (first identified in United Kingdom, B.1.1.7: N501Y), Beta (first identified in South Africa, B.1.351: K417N, E484K, N501Y) and

Gamma (first identified in Japan/Brazil, P.1: K417T, E484K, N501Y) variants were synthesized by integrated DNA technology (IDT) as gblocks and codon optimized for mammalian expression. The fragments were inserted in a pCAGGS vector using Gibson Assembly. RBD proteins were expressed in-house in Expi293F human cells (Thermo Fisher Scientific) by transfection of the cells with purified DNA and polyethylenimine (PEI). RBD from Delta variant (first identified in India, B.1.617.2: L452R, T478K), Omicron and Omicron BA.2 were purchased from Sino Biological.

## ELISAs

96-well half-area flat bottom high-bind microplates (Cultek; cat. number: 3690) were coated overnight at 4 °C with the different SARS-CoV-2 RBD recombinant viral proteins at 2 µg/ml in PBS (30 µl per well). Plates were washed with PBS 0.05% Tween 20 (PBS-T) and blocked with blocking buffer (PBS containing 1.5% Bovine serum albumin, BSA) for 2 h at room temperature (RT). Monoclonal antibodies were serially diluted (starting dilution 10 µg/ml and then 11 serial dilutions 1:4) in PBS supplemented with 0.05% Tween 20 and 1% BSA and added to the viral protein- or PBS-coated plates for 2 h at RT. After washing, plates were incubated with horseradish peroxidase (HRP)-conjugated anti-human IgG secondary antibody (Southern Biotech, 2042-05) diluted 1:4000 in PBS containing 0.05% Tween 20 and 1% BSA for 45 min at RT. Human IgG1 purified from serum of a myeloma patient (Binding Site Company, BP078) was used as a negative control. Plates were washed 5 times with PBS-T and developed with TMB substrate reagent set (BD Biosciences; cat. number: 555214) with development reaction stopped with 1 M $H_2SO_4$. Absorbance was measured at 450 nm on a microplate reader (Infinite 200 PRO, Tecan). Optical density (OD) measurement was obtained after subtracting the absorbance at 570 nm from the absorbance at 450 nm.

## Pseudovirus generation and neutralization assay

HIV reporter pseudoviruses expressing SARS-CoV-2 spike protein and Luciferase were generated as previously described[46]. pNL4-3.Luc.R-E- vector was obtained from the NIH AIDS Reagent Program[47](ARP-3418). SARS-CoV-2. SctΔ19 was generated (GeneArt) from the full protein sequence of the ancestral SARS-CoV-2 spike (UniPro.org: P0DTC2) with a deletion of the last 19 amino acids in C-terminal[48], human-codon optimized and inserted into pcDNA3.1 (+). A similar procedure was followed to generate expression plasmids for all the different variants of SARS-CoV-2 spike according to consensus data (www.outbreak.info/) (Supplementary Table 3) as well as SARS-CoV-1 spike (UniPro.org: P59594). Expi293F cells were transfected using ExpiFectamine293 Reagent (Thermo Fisher Scientific; cat. number: A14524) with pNL4-3. Luc.R-.E- and SARS-CoV-2.SctΔ19 at an 8:1 ratio, respectively. Control pseudoviruses were obtained by replacing the spike protein expression plasmid with a VSV-G protein expression plasmid. Supernatants were harvested 48 h after transfection, filtered at 0.45 mm, frozen, and titrated on HEK293T cells overexpressing wild-type human ACE-2 (Integral Molecular; cat. number: C-HA101).

The SARS-CoV-2 pseudovirus-based neutralization assay was performed in FluoroNunc F96 microwell culture plates (VWR; cat. number: 734-2017). Briefly, 200 TCID50 of pseudovirus were pre-incubated for 1 h at 37 °C with serial 1/5 dilutions of either purified mAbs, commercial neutralizing mAbs S2E12 (Proteogenix; cat. number: PTXCOV-A579) or S309 (Cell Sciences; cat. number: CPC525A). Then, $1 \times 10^4$ HEK293T/hACE2 cells treated with DEAE-Dextran (Sigma-Aldrich; cat. number: D9885) were added. Results were read after 48 h using the EnSight Multimode Plate Reader and BriteLite Plus Luciferase reagent (PerkinElmer; cat. number: 6066769). The values were normalized and the half-maximal inhibitory concentrations ($IC_{50}$) of the evaluated antibody were determined by plotting and fitting the log of the antibody concentration versus response to a 4-parameters equation in GraphPad Prism 9.0.0 (GraphPad Software).

## Virus neutralization assay

For the SARS-CoV-2 neutralization test, SARS-CoV-2 isolates were preincubated with serial 1/5 dilutions of antibody preparations for 1 h at 37 °C. Pre-incubated viruses were added to $6 \times 10^4$ Vero E6 cells (ATCC CRL-1586) per well in duplicate in 96-well plates. To control antibody-induced toxicity, Vero E6 cells were also exposed to serial dilutions of the same antibody preparations but in the absence of virus. Seventy-two hours later, viral-induced cytopathic effect was measured using the CellTiter-Glo Luciferase reagent (Promega; cat. number: G7570) and a Luminoskan Plate Reader (Thermo Fisher Scientific). The relative light units (RLU) were normalized to untreated non-infected cells (without virus), and the $IC_{50}$ (the concentration inhibiting 50% of the cytopathic effect) was calculated by plotting and fitting the log of antibody concentration vs. response to a 4-parameter equation in GraphPad Prism 9.3.1, as previously described in ref. 49.

Regarding cells, viruses, and viral titration: Vero E6 cells were cultured in Dulbecco's modified Eagle medium (Invitrogen) supplemented with 10% fetal bovine serum (FBS; Invitrogen), 100 U/ml penicillin, and 100 µg/ml streptomycin (all from Invitrogen). All viruses were isolated from clinical samples from Spain and sequenced as described in ref. 50. Genomic sequence were deposited at GISAID repository (http://gisaid.org) with the following accession IDs: D614G, EPI_ISL_510689; Alpha, EPI_ISL_1663569; Beta, EPI_ISL_1663571; Delta, EPI_ISL_3342900; BA.1.1, EPI_ISL_8151031; BA.2, EPI_ISL_11031089; BA.5, EPI_ISL_13925644; BQ.1.1, EPI_ISL_16375366. Viral stocks were propagated in Vero E6 cells for two passages and titrated in 10-fold serial dilutions to calculate the TCID50 per mL. Infection was set to achieve a 50% viral-induced cytopathic effect measured with CellTiter-Glo Luminiscent cell viability assay at 72 h.

## SARS-CoV-2 infection and prophylactic/therapeutic antibody treatment in K18-hACE2 mice

B6. Cg-Tg(K18-ACE2)2Prlmn/J (or K18-hACE2) hemizygous transgenic mice (Jackson Immunoresearch, strain: 034860) were bred, genotyped and maintained at Comparative Medicine and Bioimage Center of Catalonia (CMCiB) or purchased from Jackson Immunoresearch. For experimental infections, animals were transferred to the BSL-3 area. In all cases, room conditions were: 22 ± 2 °C, 30–70% humidity, 12 h dark/light cycle, food and water ad libitum.

For antibody prophylaxis experiments, a total of 24 adult K18-hACE2 mice (aged 6-14 weeks) were used, distributed in sex-balanced groups. One day before SARS-CoV-2 challenge, mice were administered by intraperitoneal injection with 10 mg/kg of either 17T2 mAb (n = 10, 5 males and 5 females) or an isotype control (anti-HIV-1 IgGb12) (n = 10, 5 males and 5 females). After 24 h, treated animals were anesthetized with isoflurane (FDG9623; Baxter, Deerfield, IL, USA) and challenged intranasally with 1000 TCID$_{50}$ of Omicron BA.1.1 SARS-CoV-2 isolate (EPI_ISL_8151031) diluted in PBS, or PBS only (uninfected control group). Viral challenge was performed under isoflurane anesthesia in a volume of 50 µl (25 µL/nostril). In addition, 4 mock-treated animals later named "uninfected" (2 males and 2 females) were injected intraperitoneally with PBS and challenged intranasally with PBS for comparison in the weight and clinical follow-up comparison which was performed daily. Five animals per treatment group and two per uninfected group were euthanized at 3 and 7 dpi for viral RNA quantification and pathological analyses. In all cases, euthanasia was performed under deep isoflurane anesthesia by whole blood extraction via intracardiac puncture followed by cervical dislocation. Oropharyngeal swab, lung, and nasal turbinate were collected for cell-free viral RNA quantification. Lung tissue was collected for histological and immunohistochemistry analysis. SARS-CoV-2 PCR detection, viral load

quantification, and viral titration of the samples were performed as described in ref. 33. All mice fully recovered from the infection and anesthesia procedures and no animals had to be euthanized due to Humane Endpoint Criteria considering body weight loss and clinical signs[33].

A similar procedure was carried out to assess the therapeutic efficacy of 17T2 mAb. In this set-up, transgenic mice were challenged intranasally with 1000 $TCID_{50}$ of Omicron BA.1.1 SARS-CoV-2 isolate and 24 h later were administered with 17T2 or the isotype control antibodies ($n = 8$ animals per group, 4 males and 4 females) by intra-peritoneal injection (30 mg/kg). Body weight and clinical signs were monitored daily from the antibody injection until the end of the experiment. All animals were euthanized at 5 dpi. Oropharyngeal swab and lungs were collected for cell-free viral RNA quantification. Lung tissue was collected for cell-free virus titration, and histological and immunohistochemistry analysis.

## Viral load quantification by RT-PCR

Viral RNA was quantified in several samples (lung, oropharyngeal swab, and nasal turbinate where indicated)[33]. A piece of each tissue (100 mg approximately) was collected in 1.5 ml Sarstedt tubes (Sarstedt; cat. number: 72607) containing 500 μl of DMEM medium (Thermo Fisher Scientific; cat. number: 11995065) supplemented with 1% penicillin–streptomycin (Thermo Fisher Scientific; cat. number: 10378016). A 1.5 mm Tungsten bead (QIAGEN; cat. number: 69997) was added to each tube and samples were homogenized twice at 25 Hz for 30 s using a TissueLyser II (QIAGEN; cat. number: 85300) and centrifuged for 2 min at $2000 \times g$. Supernatants were stored at −80 °C until use. RNA extraction was performed by using Viral RNA/Pathogen Nucleic Acid Isolation kit (Thermo Fisher Scientific; cat. number: A42352), optimized for a KingFisher instrument (Thermo Fisher Scientific; cat. number: 5400610), following the manufacturer's instructions. PCR amplification was based on the 2019-Novel Coronavirus Real-Time RT-PCR Diagnostic Panel guidelines and protocol developed by the American Center for Disease Control and Prevention (CDC-006-00019, v.07). Briefly, a 20 μl PCR reaction was set up containing 5 μl RNA, 1.5 μl N2 primers and probe (Integrated DNA Technologies; 2019-nCov CDC EUA Kit, cat. number: 10006770), and 10 μl GoTaq 1-Step RT-qPCR (Promega). Thermal cycling was performed at 50 °C for 15 min for reverse transcription, followed by 95 °C for 2 min and then 45 cycles of 95 °C for 10 s, 56 °C for 15 s, and 72 °C for 30 s in the Applied Biosystems 7,500 or QuantStudio5 Real-Time PCR instrument (Thermo Fisher Scientific). For absolute quantification, a standard curve was built using 1/5 serial dilutions of a SARS-CoV2 plasmid (2019-nCoV_N_Positive Control, Integrated DNA Technologies; cat. number: 10006625, used at 200 copies/μL) and run in parallel in all PCR determinations. The viral RNA of each sample was quantified in triplicate and the mean viral RNA (in copies/mL) was extrapolated from the standard curve and corrected by the corresponding dilution factor. Mouse *Gapdh* gene expression was measured in duplicate for each sample using TaqMan gene expression assay (Thermo Fisher Scientific; cat. number: Mm99999915_g1) as amplification control.

## Viral titration of replicative SARS-CoV-2 in the lungs

In the therapeutic experiment, lung tissues sampled at 5 dpi were evaluated for the presence of replicative virus by titration in Vero E6 cells[50,51]. Briefly, after tissue homogenization, each sample was diluted 2 folds and then sequentially diluted in 10-fold increments in triplicate, transferred in a 96 well plate on a Vero E6 cells monolayer, and incubated at 37 °C and 5% $CO_2$. Plates were monitored daily under the microscope, and, at 5 dpi, wells were evaluated for the presence of cytopathic effects. The amount of infectious virus was calculated by determining the TCID50 using the Reed–Muench method.

## Histopathology and SARS-CoV-2 immunohistochemistry

Lungs from mice were collected at the indicated time, fixed by immersion in 10% buffered formalin and embedded into paraffin blocks. The histopathological analysis was performed on slides stained with hematoxylin/eosin and examined by optical microscopy. A semi-quantitative score based on the level of broncho-interstitial pneumonia (0 = No lesion; 1 = Mild, 2-Moderate or 3 = Severe lesion) was established based on previous classifications[52,53]. SARS-CoV-2 nucleo-protein was detected by immunohistochemistry (IHC) using the rabbit monoclonal antibody (40143-R019, Sino Biological) at a 1:15000 dilution. For immunolabelling visualization, the EnVision+ System linked to horseradish peroxidase (HRP, Agilent-Dako) and 3,3'-diamino-benzidine (DAB) were used. The amount of viral antigen in tissues was semi-quantitatively scored (0 = no antigen, 1 = low and multifocal antigen, 2 = moderate and multifocal antigen, 3 = high and diffuse antigen) following previously published classifications[52,53].

## Determination of binding kinetics by surface plasmon resonance

Binding kinetics and affinity of 17T2 mAb for RBD were evaluated by surface plasmon resonance on a BIAcore T100 instrument (Cytiva) with a running buffer composed of 10 mM HEPES at pH 7.2, 150 mM NaCl and 0.05% Tween 20. The assay format involved antibody capture on a Series S CM5 chip. Briefly, amine coupling was used to create a human IgG capture surface (anti-human Fc mAb) following instructions provided with the Cytiva human IgG capture kit. 17T2 mAb was captured on flow cell 2, leaving flow cell 1 as a subtractive reference. Capture levels of IgG were targeted between 100 and 200 resonance units, after which serial dilution of RBD was flowed over immobilized IgG (50 μL/min for 2 min) and allowed to dissociate up to 30 min. The capture surface was regenerated with a 60-s injection of 3 M $MgCl_2$ (50 μL/min for 1 min). A 2-fold concentration series of each RBD variants ranging from 2 to 0.25 nM was used to analyze binding to 17T2. All sensorgrams were analyzed using a 1:1 Langmuir binding model with software supplied by the manufacturer to calculate the kinetics and binding constants. Where no decay in the binding signal was observed during the time allowed for dissociation, $K_D$ based on the kd limit was determined by the "5% rule"[54].

## De-glycosylation of monoclonal antibodies

Following the manufacturer's instructions for PNGase F (Promega Inc.; cat. number: V483A) treatment, 200 μg of 17T2 mAb in 50 mM ammonium bicarbornate (pH 7.4) was combined with 50 μL of Glyco Buffer 10X and water to make up a 500 μL total reaction volume. The mixture was incubated at 37 °C overnight without or with 20 μL of PNGase F. Control analysis of the mAb de-glycosylation was performed by gel-shift on SDS-PAGE. The mAb preparation was concentrated through a 50 kDa Amicon centrifugal filter to remove the PNGase enzyme (EMD Millipore) and dialyzed in PBS 1X for further experiments.

## Cryo-electron microscopy sample preparation and data acquisition

B.1.1.529 (BA.1) S1 + S2 trimer-His Recombinant Protein (Sino Biological; cat. number: 40589-V08H26) was reconstituted in sterile water (100 μL) to prepare a stock solution (1 mg/mL). Buffer exchange to 10 mM Tris pH 7.6 was performed twice through Zeba Spin Desalting Columns (Thermo Fisher Scientific; cat. number: 899882). BA.1 spike was mixed with the Fab 17T2 (1:1.5 molar ratio of spike monomer: Fab, i.e., 0.59:0.71 mg/mL, respectively) and 3 μL were kept 5 min at RT until their application onto glow-discharged holey carbon grids (Quantifoil, Au 300 mesh, R 0.6/1; cat. number: 4N1-C11nAu30). The grids were blotted and then plunged into liquid ethane using a FEI Vitrobot Mark IV at 20 °C and 95% relative humidity. Data were collected at a FEI Talos Arctica electron microscope operated at 200 kV and equipped with a

Falcon III electron detector. A total of 9,615 movies (Supplementary Fig. 4a) were recorded at a defocus range of −1 μm to −2.5 μm with a pixel size of 0.855 Å; exposure time was 40 s, with a total exposure dose of 32 e/Å2 over 60 frames.

## Image processing

All image processing steps were performed inside Scipion[55]. We used Scipion 3.0 in order to easily combine different software suits in the analysis workflows of cryo-EM data: movie frames were aligned using Relion's implementation of the UCSF MotionCor2 program[56,57]. The contrast transfer function (CTF) of the micrographs was estimated using GCTF[58]. Movies were then automatically picked using Gautomatch. Following the application of the Scipion picking consensus protocol, 1,203,207 particles were extracted. 2D classification was performed in cryoSPARC[59] and 124,570 particles were selected (Supplementary Fig. 4b). The cryoSPARC initial model protocol was then used to generate and classify the particles into 2 classes, without imposing symmetry (Supplementary Fig. 4c). Class 1 contained low-quality particles (16,866) and class 2 contained a dataset with the highest number of particles with high quality particles (107,704 particles). The highest dataset was selected to perform a further classification yielding indistinguishable classes in which all the RBDs were in up conformation. The dataset with the highest number of particles (107,704 particles) was refined using non-uniform refinement in cryoSPARC with no symmetry application (Supplementary Fig. 4d), to overall resolution of 3.46 Å based on the gold-standard (FSC = 0.143) criterion (Supplementary Fig. 4f). The resulting map was sharpened with DeepEMhancer[60]. The coordinates of the SARS-CoV-2 spike in PDB ID: 7Y9S [https://doi.org/10.2210/pdb7Y9S/pdb] were used as an initial model for fitting the cryo-EM map. Output from AlphaFold 2.0 modeling was used as an initial model for 17T2 Fab. Due to the higher flexibility in the RBD-antibody region and consequently the lower resolution in this area, local refinements with CryoSPARC were performed, using a mask encompassing the RBD-Fab region (Supplementary Fig. 4e). The final resolution was 4.41 Å map, which allowed for increased definition in this region (Supplementary Fig. 4g). However, the side chains were not fully resolved.

## Model building and refinement

Iterative manual model building was carried out in Coot[61] and refinement in Phenix[62] and Refmac5 in CCP-EM[63]. The validation of the model was done with Molprobity[64] (Supplementary Table 5) sofware integrated in Phenix suite. UCSF Chimera and ChimeraX were used for map fitting and manipulation[65]. PDBePISA[66] and PDBsum[67] servers were used for interaction analysis (residues involved and area of interaction) between 17T2 Fab and RBD.

## Statistical analysis

All figures were generated in GraphPad Prism 9.0.0. Statistical analyses were performed using R v4.1.1. Unpaired datasets were analyzed using a Kruskal-Wallis with Dunn's correction for multiple testing. Histopathological and IHC scores were compared using the Generalized Pearson Chi-Squared Test for ordinal data. Viral load comparisons were analyzed using a Petro-Prentice generalized Wilcoxon test. Viral Titration was compared using a Peto-Peto Left-censored 2-sample test. All performed tests were two-sided.

## Reporting summary

Further information on research design is available in the Nature Portfolio Reporting Summary linked to this article.

## Data availability

The cryo-EM data generated in this study have been deposited in the Electron Microscopy Data Bank under accession codes EMD-16453 (https://www.emdataresource.org/EMD-16453) for SARS-CoV-2 spike trimer in complex with three 17T2 Fabs and EMD-16473 (https://www.emdataresource.org/EMD-16473) for RBD/17T2 Fab. The associated atomic models generated in this study have been deposited in the Protein Data Bank under accession code 8C89 (https://www.wwpdb.org/pdb?id=pdb_00008c89). Further information and requests for resources and reagents should be directed to and will be fulfilled by the corresponding author, Giuliana Magri (gmagri@imim.es). All other data are available in the article and its Supplementary files or from the corresponding author upon request. Source data are provided with this paper.

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

## Acknowledgements

We acknowledge access to the cryo-EM CNB-CSIC facility in the context of the CRIOMECORR project (ESFRI-2019-01-CSIC-16) and we thank the staff of the Protein Technology Unity (CRG) for their help in protein production. We thank Antonio Barreiro and Laura Ferrer from HIPRA, S.A. for the BA.2.86 spike expression plasmid. This study was supported by the COVID-19 call grant from Generalitat de Catalunya, Department of Health (to G.M.), grant Miguel Servet research program (to G.M.), and partially funded by the crowdfunding initiative #joemcorono and the Fundació Glòria Soler (to J.B.). A.P.-G. was supported by a predoctoral grant from Generalitat de Catalunya and Fons Social Europeu (2022 FI_B 00698).

## Author contributions

L.D.C.-M. produced the monoclonal antibodies, designed and performed experiments, analyzed and discussed data, and reviewed and edited the manuscript. S.T.V. produced the monoclonal antibodies, designed and performed experiments, and discussed data. N.R.M. performed SARS-CoV-2 antigens and mAbs expression and purification and carried out SPR experiments. B.T. carried out neutralization assays and in vivo experiments with K18-hACE-2 mice, analyzed and interpreted the data and contributed to the writing of the manuscript. E.P. and S.M. carried out neutralization assays. A.P.-G. carried out in vivo experiments with K18-hACE-2 mice, analyzed and interpreted the data and contributed to the writing of the manuscript. F.T.-F. and N.I.-U. carried out in vivo experiments with K18-hACE-2 mice. F.A. carried out neutralization assays and in vivo experiments. V.U. performed statistical analyses. D.R-R. carried out authentic virus neutralization assays and titrations. E.R.-M. performed VL quantification. E.B. performed VL quantification and analyzed and interpreted the data. J.V.-A. carried out immunohistochemistry and histology experiments and analyzed and interpreted the data. M.P. carried out immunohistochemistry and histology experiments. J.S. carried out immunohistochemistry and histology experiments and analyzed and interpreted the data. A.M. carried out computational aspects of image processing and structure determination, interpreted and analyzed the cryo-EM structures and binding analyses and carried out experimental aspects of cryo-EM. M.T.B.-C. collected cryo-EM data. D.C. carried out computational aspects of image processing and structure determination. C.S. interpreted and analyzed the cryo-EM structures and binding analyses. R.A. carried out experimental aspects of cryo-EM, interpreted and analyzed the cryo-EM structures and binding analyses and wrote the manuscript. C.C. designed and performed experiments, carried out SPR experiments, discussed data, and reviewed the manuscript. J.B analyzed and interpreted the data and wrote the manuscript. G.M. conceived and supervised the study, produced the monoclonal antibodies, analyzed and interpreted data, and wrote the manuscript.

## Competing interests

Unrelated to the submitted work, J.B. is the founder and shareholder of AlbaJuna Therapeutics, S.L.; J.B. reports institutional grants from Grifols, HIPRA, NESAPOR Europe, and MSD. G.M., S.T.V., L.D.C.-M., C.C., J.B., B.T., E.P. (co-authors of this work) are inventors on a patent application (EP22382940) related to this work. The authors declare no other competing interests.

## Additional information

¹Translational Clinical Research Program, Hospital del Mar Research Institute (IMIM), Barcelona, Spain. ²IrsiCaixa AIDS Research Institute, Hospital Germans Trias I Pujol, Campus Can Ruti, Badalona, Spain. ³Centro Nacional de Biotecnología (CNB-CSIC), Madrid, Spain. ⁴Centre for Genomic Regulation (CRG), The Barcelona Institute of Science and Technology, Barcelona, Spain. ⁵CIBERINFEC, ISCIII, Madrid, Spain. ⁶Germans Trias i Pujol Research Institute (IGTP), Can Ruti Campus, Badalona, Spain. ⁷Unitat Mixta d'Investigació IRTA-UAB en Sanitat Animal. Centre de Recerca en Sanitat Animal (CReSA), Campus de la Universitat Autònoma de Barcelona (UAB), Bellaterra, Spain. ⁸IRTA. Programa de Sanitat Animal. Centre de Recerca en Sanitat Animal (CReSA), Campus de la Universitat Autònoma de Barcelona (UAB), Bellaterra, Spain. ⁹Departament de Sanitat i Anatomia Animals, Facultat de Veterinària, Universitat Autònoma de Barcelona (UAB), Bellaterra, Spain. ¹⁰Infectious Diseases and Immunity, Faculty of Medicine, University of Vic-Central University of Catalonia (UVic-UCC), Barcelona, Spain. ¹¹Present address: Division of Immunology, Department of Medical Biochemistry and Biophysics, Karolinska Institute, Stockholm, Sweden. ¹²Present address: IRTA. Programa de Sanitat Animal. Centre de Recerca en Sanitat Animal (CReSA), Campus de la Universitat Autònoma de Barcelona (UAB), Bellaterra, Spain. ¹³Present address: Immunology Unit, Department of Biomedical Sciences, Faculty of Medicine and Health Sciences, University of Barcelona, Barcelona, Spain. ¹⁴These authors contributed equally: Leire de Campos-Mata, Benjamin Trinité, Andrea Modrego. ¹⁵These authors jointly supervised this work: Carlo Carolis, Rocío Arranz, Julià Blanco, Giuliana Magri. ✉e-mail: carlo.carolis@crg.eu; rarranz@cnb.csic.es; jblanco@irsicaixa.es; gmagri@imim.es

