## [Peer Review File · Nature Communications]

A Monoclonal Antibody Targeting a Large Surface of the Receptor Binding Motif Shows Pan-Neutralizing SARS-CoV-2 ActivityREVIEWER COMMENTS

Reviewer #1 (Remarks to the Author):

Monoclonal antibodies specific to the spike protein from SARS-CoV-2 have been used as therapeutic agents to treat COVID-19. However, the treatment with the commercially available anti-Spike mAbs are not effective against the variants of concern (VoC) and in particular against the omicron sub-variants. This study reports the identification of mAb generated from B lymphocytes of convalescent individuals. Both the original RBD-specific B cells were well characterized and the heavy and light chain sequenced. Importantly, one of the anti-spike mAbs (17T2 mAb) was shown to bind and to possess a broadly neutralizing activity against spike of various VoCs, as well as a prophylactic activity in vivo for the omicron variant. Finally, cryo-microscopy analysis was performed to understand the interaction the 17T2 mAb with the receptor binding domain of spike, and to further understand the broadly neutralizing activity nature of the 17T2 mAb. This is an important finding and may lead to develop a mAb that may be useful to treat patients infected with SARS-CoV2 variants. The manuscript is straightforward, however, the functional activity of the 17T2 mAb needs a better characterization using neutralizing assays with the real virus and an experimental design that resemble the therapeutic use in humans.

Concerns to be addressed:

- 1) Ideally, the broad neutralizing activity of the 17T2 mAb should be tested with the ancestral SARS-CoV2 and the VoCs viruses. It is justified to perform the initial screening of neutralizing mAbs with the pseudovirus expressing the different Spikes from VoCs. However the final characterization and comparison with existing mAbs (S2E12 and S309) possessing broadly neutralizing activity should be performed with the real SARS-CoV2 ancestral and VoCs viruses.
- 2) Neutralizing mAbs can be used prophylactically, however the main use is therapeutic. The in vivo experiments should reflect its main use. Mice should be infected, wait for few days and then treated with the mAbs.

3) It seems that a dose of 10 mg/Kg is very high. How this reflects to the dose of anti-SARS-CoV2 mAbs used in human treatment? This maybe an important limitation to develop the 17T2 mAb as a reagent to treat humans.

4) It is important to test the therapeutic effect of the 17T2 mAb against the ancestral and few other variants. Ideally, these studies should be compared to the S2E12 antibodies that seem to have much higher neutralizing activity to the ancestral virus and also higher activity when compared to the pre-omicron VoCs.

Finally, the results are difficult to follow, they need to be better described, since Nature Communication aims a broader readership. The use of many anacronyms that often do not correspond to the ones shown in Figures makes difficult to follow the results description. Some results, in particular of the physical-chemical assays, shown both in Figures and tables are poorly explained. The figure legends are often not sufficient, so the reader can understand the experiments not having to go back to the main text.

Reviewer #2 (Remarks to the Author):

De Campos-Mata and colleagues describe a new broadly neutralizing antibody (bNAb) against SARS-CoV-2. This antibody, named 17T2, was isolated from a convalescent individual after infection in the first wave of the COVID-19 pandemic. The authors sorted 380 peripheral blood B cells reactive to the SARS-CoV-2 spike receptor binding domain (RBD), performed single cell cloning, and produced five monoclonal antibodies. These were screened for broad reactivity against SARS-CoV-2 variants by ELISA and pseudoparticle neutralization assays. 17T2 was the broadest and most potent of the five antibodies, neutralizing 10 out of 10 tested SARS-CoV-2 strains including the Omicron variants BA.1, BA.2, BA.4/5, and BQ.1.1 at IC₅₀ values of 2 – 38 ng/ml. The authors then tested in vivo prophylactic activity of 17T2 against the BA.1.1 variant using K18-hACE2 transgenic mice. Animals treated with 17T2 prior to infection showed lower viral loads in the oropharynx, lung, and nasal turbinate as compared to animals treated with an isotype control antibody. 17T2 treatment also protected from histopathological lesions in the lung. Finally, cryo-EM analysis demonstrated that 17T2 binds to the spike protein in an RBD-up position via a

relatively large surface area overlapping with the receptor binding motif. The RBD-up binding mode, together with 17T2's usage of the VH1-58 gene segment, is typical for a public class of SARS-CoV-2 antibodies. However, the relatively large surface area occupied by 17T2 may possibly explain its broad activity.

The data are clearly presented and the methods are well described. Although the authors demonstrate that 17T2 is a well neutralizing SARS-CoV-2 candidate, other antibodies are available with similar or superior activities. It would be important to see, if this antibody also neutralizes the newer variants including XBB1.5 and BF.7.

Specific points

1. This referee is aware that new SARS-CoV-2 variants are constantly emerging and it is challenging to be up to date with neutralization data at the time of publication. However, in view of the fact that this study basically revolves around only one antibody, the pseudovirus panel needs be expanded to include more variants such as the recently evolved XBB.1.5 variant.
2. Data points for neutralization curves and IC50 values in Figure 1 are means \pm SD of duplicate samples. For $n=2$, it is better to show the two values than to calculate an SD. This n number is also quite small.
3. In Supplementary Figure 2, why does the negative control show a positive ELISA signal for the Delta variant?
4. The order of panels in Figure 1 is confusing. For the first experiment, IC50 values are shown first and curves are shown second. For the second experiment, curves are shown first and values second. This referee would suggest to show neutralization curves and IC50 values in the same order for both experiments to improve clarity.

Reviewer #3 (Remarks to the Author):

Overall, a very clear paper characterizing a mAb (17T2) isolated from a convalescent COVID-19 person infected with the original strain. The authors find that this antibody has broad neutralizing effects for all variants tested and is specific to Sars Cov2. Importantly, it has potent neutralizing activity against a wide range of virus strains known to evade previous mAbs raised against the original virus strain. The authors go on to show that this mAb is effective in protecting against SarsCOV2 BA1.1 in transgenic mice. Finally, the authors structurally characterize the binding interface of the mAb with the receptor binding domain (RBD) of the spike trimer. They identify a series of residues involved in hydrogen bonding and 1 salt bridge to stabilize the interface. They show that it is a larger interaction interface than those found for other mAbs and suggest that may be why 17T2 is more resilient to spike mutations.

The cryoEM workflow is straight forward. I would recommend moving the methods describing the focused refinement used to obtain a higher resolution structure of the interface from the “Model building and refinement” section to the “Image processing” section.

In looking at the PDB/EMDB validation reports a couple of things are flagged.

- 1) The reported resolution of 3.46 differs by more than 10% from the resolution estimate based on the FSC curve calculated from the deposited half maps (Validation report p. 11 section 4.2). Could the authors comment on this discrepancy?
- 2) Why in the validation report for EMD-16473, does the FSC never reach 0? When it does in Supplementary Figure 3G in the manuscript?
- 3) The clash score is high for the associated PDB (25.7). The authors mention that they don't observe side-chain density; however, they don't indicate what kind of restraints were used in the model refinement procedure? If something like Isolde were used with adaptive distance restraints would the clash score improve?

Supplementary table 5 could do with a legend describing some of the acronyms used in the table (d99, CaBLAM etc) What is being reported in the Model vs Data section with the various cross-correlations? How were these calculated? What does cc peaks and cc volume

refer to?

A recently published study in PLOS Biology characterized another neutralizing mAb isolated from convalescent patient infected with the prototype strain (Meng et al., 2023). This study applied similar types of analyses including neutralizing capability, therapeutic efficacy in transgenic mice and structural biology. The discussion section could be expanded to describe how their results compares/contrasts to those presented in this new study.

<https://journals.plos.org/plospathogens/article?id=10.1371/journal.ppat.1011085>

One more final suggestion on how to improve the clarity of the structural figures. Figure 2E, it is hard to see the lines of the H bond network. I would suggest removing the lines and instead leave the residues involved labelled which form the key interactions at the interface. If you color the sticks according to heteroatom, it will also be easier for the reader to identify the residues you refer to.

When you describe the increased interface area for the binding site for 17T2, it would be helpful to include quantitative measurements for this, for example buried surface area in PISA.

<https://www.ebi.ac.uk/pdbe/pisa/>

Text in black shows the Reviewers' comments, whereas **text in blue** shows the Authors' response to the Reviewers' comments.

Changes in the Revised Manuscript are highlighted in **turquoise**.

Reviewer #1 (Remarks to the Author):

Monoclonal antibodies specific to the spike protein from SARS-CoV-2 have been used as therapeutic agents to treat COVID-19. However, the treatment with the commercially available anti-Spike mAbs are not effective against the variants of concern (VoC) and in particular against the omicron sub-variants. This study reports the identification of mAb generated from B lymphocytes of convalescent individuals. Both the original RBD-specific B cells were well characterized and the heavy and light chain sequenced. Importantly, one of the anti-spike mAbs (17T2 mAb) was shown to bind and to possess a broadly neutralizing activity against spike of various VoCs, as well as a prophylactic activity *in vivo* for the omicron variant. Finally, cryo-microscopy analysis was performed to understand the interaction the 17T2 mAb with the receptor binding domain of spike, and to further understand the broadly neutralizing activity nature of the 17T2 mAb. This is an important finding and may lead to develop a mAb that may be useful to treat patients infected with SARS-CoV2 variants. The manuscript is straightforward, however, the functional activity of the 17T2 mAb needs a better characterization using neutralizing assays with the real virus and an experimental design that resemble the therapeutic use in humans.

We thank the Reviewer for his/her words of appreciation for our work and we agree with the Reviewer that we should improve the characterization of 17T2 mAb.

Concerns to be addressed:

- 1) Ideally, the broad neutralizing activity of the 17T2 mAb should be tested with the ancestral SARS-CoV2 and the VoCs viruses. It is justified to perform the initial screening of neutralizing mAbs with the pseudovirus expressing the different Spikes from VoCs. However, the final characterization and comparison with existing mAbs (S2E12 and S309) possessing broadly neutralizing activity should be performed with the real SARS-CoV2 ancestral and VoCs viruses.

We agree with the Reviewer and Supplementary Figure 3 in the Revised Manuscript now shows the neutralizing activity of 17T2 mAb, S2E12, and S309 in *in vitro* assays using SARS-CoV-2 D614G, Alpha, Beta, Delta and Omicron BA.1.1, BA.2, BA.5, and BQ.1.1 authentic viruses. This new data set confirms the broad inhibitory activity of 17T2 monoclonal antibody, as well as its increased resilience against Omicron variants compared to the structurally related antibody S2E12. The Results and Material & Methods sections have been updated accordingly (pages 6 and pages 15 in the Revised Manuscript).

- 2) Neutralizing mAbs can be used prophylactically, however the main use is therapeutic. The *in vivo* experiments should reflect its main use. Mice should be infected, wait for few days and then treated with the mAbs.

In response to the request from Reviewer 1, we expanded the characterization of 17T2 antibody by conducting a new *in vivo* experiment to assess its therapeutic activity. We chose the K18 hACE2 mouse model of BA.1.1 infection, the same model used for assessing prophylactic activity (**Figure 2** in the manuscript), for comparative purposes. We followed a protocol similar to that of Case et al, 2022, doi:10.1038/s41467-022-31615-7, where antibody treatment was administered one day after virus challenge. This protocol aligns with the common practice in this model due to the rapid progression of infection.

Specifically, K18 hACE2 mice were intranasally challenged with 10^3 TCID₅₀ equivalents of BA.1.1. One day after, 30 mg/kg of 17T2 mAb (n=8) or IgGb12 isotype control (n=8) were administered via intraperitoneal injection. At day 5 post-infection, we collected oropharyngeal swabs for viral load analysis and examined lung tissues for viral load, histology, and immunohistochemistry (IHC) analysis using the same procedures as in the prophylactic setting (see Materials and Methods section in the **Revised Manuscript**). As viral load determinations by qPCR captured both infectious and non-infectious viral genomes, we also checked viral infectivity using lung tissues from infected mice to confirm the impact of 17T2 mAb on infectious viral titers *in vivo*.

The use of a higher dose of antibody (30 mg/kg) in the therapeutic experiment compared to the prophylactic set-up is justified by the fact that 17T2 mAb is a human antibody that has not been Fc-engineered for this specific experimental setting. Indeed, in the study by Case et al, 2022, the therapeutic activity of S309 in the same mouse model was demonstrated using 30 mg/kg of S309 antibody, which in addition was modified to increase interaction with neonatal Fc receptors. Similarly, *in vivo* activity of S2E12 was tested in Syrian Golden hamsters using a hamster-Fc engineered antibody (Tortorici et al 2020, doi:10.1126/science.abe3354) to better recapitulate Fc-dependent antibody functions relevant for *in vivo* activity.

The results obtained indicate that the administration of 17T2 mAb significantly reduced the viral load in oropharyngeal swabs 5 days post infection, in comparison to mice infected and treated with an isotype control antibody. While the effect on viral load reduction in the lungs was more modest, the administration of 17T2 mAb drastically reduced viral infectivity and, more importantly, tissue lesions in lungs, confirming its post-exposure therapeutic activity. All data has been summarized in the new **Figure 3** of the **Revised Manuscript**. The Results, Legends and Material & Methods sections have been updated accordingly (page 7, 16 and 27 in the **Revised Manuscript**).

- 3) It seems that a dose of 10 mg/kg is very high. How this reflects to the dose of anti-SARS-CoV2 mAbs used in human treatment? This maybe an important limitation to develop the 17T2 mAb as a reagent to treat humans.

We decided to use 10 mg/kg as prophylactic dose in our *in vivo* experiment based on previous literature. As summarized in Table 1 of Hwang et al.'s paper (Hwang et al, 2022, doi:10.1186/s12929-021-00784-w), most of approved SARS-CoV-2 neutralizing mAbs that were tested for prophylactic activity *in vivo* were administered at a concentration of 10 mg/kg or higher. The *in vivo* prophylactic activity of REGN-COV2 (casirivimab and imdelivimab) was tested in rhesus macaques and golden hamsters and REGN-COV2 was reported to be effective when administrated prophylactically at 50 mg/kg dosage (Baum

et al, 2020, doi:10.1126/science.abd0831). AZD7442 (tixagevimab and cilgavimab) prophylactic activity was tested in hACE2 transgenic mice at 10 mg/kg and in Rhesus macaques at 50 mg/kg (Zost et al, 2020, doi:10.1038/s41586-020-2548-6). LyCOV016 (Etesevimab, CB6 JS016) prophylactic activity was evaluated in rhesus macaques at 50 mg/kg (Shi et al, 2020, doi:10.1038/s41586-020-2381-y). Bamlanivimab (Ly-CoV555) is one of the few mAbs that was shown to be effective in reducing viral replication (SARS-CoV2 ancestral strain) in a rhesus macaque challenge model with a prophylactic dose of as low as 2.5 mg/kg (Jones et al, 2021, doi:10.1126/scitranslmed.abf1906).

The prophylactic activity of Sotrovimab (S309) and AZD7442 against Omicron variants was recently tested *in vivo* at 10 mg/kg in K18-hACE2 transgenic mice (Case et al, 2022, doi:10.1038/s41467-022-31615-7). In other recent publications, *in vivo* protection against Omicron and other SARS-CoV2 variants was assessed in K18-hACE2 transgenic mice at the same dose we have used (10 mg/kg) (Ju et al, 2023, doi:10.1038/s41590-023-01449-6; Du et al, 2022, doi:10.1126/sciimmunol.abp9312; Hastie et al, 2023, doi:10.1016/j.celrep.2023.112421).

During the pandemic, emergency use authorization has been granted to several monoclonal antibodies (mAbs) for use in humans. For instance, sotrovimab was recommended at a dosage of 500 mg, given as a single intravenous (IV) infusion. Bamlanivimab and etesevimab were administered together, with bamlanivimab at 700 mg and etesevimab at 1400 mg. Similarly, casirivimab and imdevimab were authorized for emergency use in combination at a dosage of 600 mg each. Notably, bebtelovimab had the lowest recommended dosage among the mAbs, with a dose of 175 mg for adult humans.

In summary, while we agree with the Reviewer that conducting testing of our monoclonal antibody (mAb) at lower doses (2 or 5 mg/kg) would have provided additional information, our choice of experimental setting for *in vivo* experiments aligns with previous studies in the field and matches the recommended dosages for emergency use of mAbs in humans. Furthermore, the dosage used in a mouse model may not accurately reflect the dosage administered to humans. This distinction arises from the fact that 17T2 mAb is a human antibody that has not been engineered for Fc functionality in mice.

- 1) It is important to test the therapeutic effect of the 17T2 mAb against the ancestral and few other variants. Ideally, these studies should be compared to the S2E12 antibodies that seem to have much higher neutralizing activity to the ancestral virus and also higher activity when compared to the pre-omicron VoCs.

We acknowledge the importance of testing the effect of 17T2 mAb *in vivo* against the ancestral and other variants, as suggested by the Reviewer. However, due to limited funds, we have decided to exclusively use the BA.1.1 isolate from the more recent Omicron variant family for our therapeutic *in vivo* experiment. This isolate had already been characterized in previous *in vivo* studies in the Original Manuscript. For the same practical reasons, we opted to exclude the comparison of *in vivo* activity between 17T2 mAb and S2E12. We are confident that the *in vitro* neutralization assays, conducted with both pseudovirus and authentic virus, are sufficient to expand the characterization of 17T2 against all the variants and allow for comparisons with other relevant neutralizing

monoclonal antibodies, such as S309 and the structurally similar S2E12 (**Figure 1 and Supplementary Figure 3 in the Revised Manuscript**).

Finally, the results are difficult to follow, they need to be better described, since Nature Communication aims a broader readership. The use of many acronyms that often do not correspond to the ones shown in Figures makes difficult to follow the results description. Some results, in particular of the physical-chemical assays, shown both in Figures and tables are poorly explained. The figure legends are often not sufficient, so the reader can understand the experiments not having to go back to the main text.

We have incorporated the Reviewer's feedback, resulting in a **Revised Manuscript** with fewer acronyms throughout the text. We have also enhanced the clarity of the figure legends and tables. Furthermore, we have carefully revised the text to ensure that the acronyms mentioned in the main text align with those used in the Figures and Figure legends, thereby improving the manuscript's overall clarity and cohesiveness.

Reviewer #2 (Remarks to the Author):

De Campos-Mata and colleagues describe a new broadly neutralizing antibody (bNAb) against SARS-CoV-2. This antibody, named 17T2, was isolated from a convalescent individual after infection in the first wave of the COVID-19 pandemic. The authors sorted 380 peripheral blood B cells reactive to the SARS-CoV-2 spike receptor binding domain (RBD), performed single cell cloning, and produced five monoclonal antibodies. These were screened for broad reactivity against SARS-CoV-2 variants by ELISA and pseudoparticle neutralization assays. 17T2 was the broadest and most potent of the five antibodies, neutralizing 10 out of 10 tested SARS-CoV-2 strains including the Omicron variants BA.1, BA.2, BA.4/5, and BQ.1.1 at IC50 values of 2 – 38 ng/ml. The authors then tested *in vivo* prophylactic activity of 17T2 against the BA.1.1 variant using K18-hACE2 transgenic mice. Animals treated with 17T2 prior to infection showed lower viral loads in the oropharynx, lung, and nasal turbinate as compared to animals treated with an isotype control antibody. 17T2 treatment also protected from histopathological lesions in the lung. Finally, cryo-EM analysis demonstrated that 17T2 binds to the spike protein in an RBD-up position via a relatively large surface area overlapping with the receptor binding motif. The RBD-up binding mode, together with 17T2's usage of the VH1-58 gene segment, is typical for a public class of SARS-CoV-2 antibodies. However, the relatively large surface area occupied by 17T2 may possibly explain its broad activity. The data are clearly presented and the methods are well described. Although the authors demonstrate that 17T2 is a well neutralizing SARS-CoV-2 candidate, other antibodies are available with similar or superior activities. It would be important to see, if this antibody also neutralizes the newer variants including XBB1.5 and BF.7.

We would like to express our appreciation to the Reviewer for their encouraging remarks. We also agree with the Reviewer's suggestion to test the neutralization activity of 17T2 against the latest variants.

After careful consideration, we have made the decision to focus our analysis specifically on the following newest variants: XBB.1.5, XBB.1.16, EG5.1 and BA.2.86. This decision is based on information provided by the World Health Organization (WHO) (source: <https://www.who.int/activities/tracking-SARS-CoV-2-variants>), which designates

XBB.1.5, XBB.1.16 and EG5.1 as current Variants of Interest (VOIs). Additionally, we have decided to test the neutralizing activity of 17T2 mAb against the recently identified BA.2-derived BA.2.86 variant, as it has raised global concerns due to its significant mutational load.

On the other hand, we have decided not to test Omicron BF.7 because it is currently detected at very low levels (<https://www.ecdc.europa.eu/en/covid-19/variants-concern>). It is worth noting that BF.7 is an Omicron subvariant of BA.5 and carries an additional mutation (R346T) in a region that lies outside the contact area of 17T2 mAb with the receptor-binding domain (RBD).

Specific points:

1. This referee is aware that new SARS-CoV-2 variants are constantly emerging and it is challenging to be up to date with neutralization data at the time of publication. However, in view of the fact that this study basically revolves around only one antibody, the pseudovirus panel needs be expanded to include more variants such as the recently evolved XBB.1.5 variant.

We agree with the Reviewer and **Revised Figure 1**, panel **C**, **D** and **E** now shows the neutralizing activity of 17T2 mAb, S2E12, and S309 in *in vitro* assays using SARS-CoV-2 XBB.1.5, XBB.1.16, EG.5.1, and BA.2.86 pseudovirus. The Abstract, Introduction, Material & Methods, Results and Discussion sections have been updated accordingly (pages 3-5, 9-10, **Figure 1**, **Figure 4** and **Supplementary Table 3** and **4** in the **Revised Manuscript**). This analysis reveals that 17T2 mAb is still able to neutralize the currently circulating VOIs as well as BA.2.86, the latter categorized by the WHO as “under monitoring”. However, the neutralizing activity of 17T2 mAb against these newest variants is reduced compared to pre-Omicron and other Omicron variants, with an IC50 value ranging from 0.42 µg/ml for XBB.1.16 to 1.18 µg/ml for EG.5.1. We have hypothesized in the discussion section of the **Revised Manuscript** the reasons for the diminished neutralizing activity of 17T2 against these newest variants compared to related variants (highlighted in turquoise, pages 9-10). Despite the observed reduction, 17T2 mAb still retains neutralizing activity against all variants tested, probably due to the larger contact area of interaction, distinguishing 17T2 from other previously described neutralizing antibodies (references 41-43 in the **Revised Manuscript**), including structurally similar antibodies such as S2E12. Therefore, the main message of our manuscript remains unchanged.

2. Data points for neutralization curves and IC50 values in Figure 1 are means ± SD of duplicate samples. For n=2, it is better to show the two values than to calculate an SD. This n number is also quite small.

We thank the Reviewer for this remark. IC50 values are calculated from curves generated from duplicate samples of serial dilutions. However, neutralization experiments using HIV reporter pseudoviruses expressing different SARS-CoV-2 spike proteins were performed several times with highly consistent results. This information is now reported in the **Revised Figure Legend 1**. We also fully agree that showing all values in the figures is the best option. Therefore, all Figure panels showing neutralization curves have been modified in the **Revised Manuscript**. Figure legends

have been modified accordingly by adding the following sentence: Duplicate values corresponding to a representative experiment out of at least two are shown.

3. In Supplementary Figure 2, why does the negative control show a positive ELISA signal for the Delta variant?

We would like to express our gratitude to the Reviewer for bringing this matter to our attention. The higher optical density (OD) values observed for the negative control at higher concentrations were a result of an extended exposure time to TMB before stopping the reaction. To address this issue, we conducted additional ELISA experiments specifically for Delta, Omicron BA.1, and Omicron BA.2. **The revised Supplementary Figure 2** now incorporates these new data sets.

4. The order of panels in Figure 1 is confusing. For the first experiment, IC50 values are shown first and curves are shown second. For the second experiment, curves are shown first and values second. This referee would suggest neutralization curves and IC50 values in the same order for both experiments to improve clarity.

We agree with the Reviewer, and in the **Revised Manuscript** we have changed the order of the panels to improve clarity (revised **Figure 1**).

Reviewer #3 (Remarks to the Author):

Overall, a very clear paper characterizing a mAb (17T2) isolated from a convalescent COVID-19 person infected with the original strain. The authors find that this antibody has broad neutralizing effects for all variants tested and is specific to Sars Cov2. Importantly, it has potent neutralizing activity against a wide range of virus strains known to evade previous mAbs raised against the original virus strain. The authors go on to show that this mAb is effective in protecting against SarsCOV2 BA1.1 in transgenic mice. Finally, the authors structurally characterize the binding interface of the mAb with the receptor binding domain (RBD) of the spike trimer. They identify a series of residues involved in hydrogen bonding and 1 salt bridge to stabilize the interface. They show that it is a larger interaction interface than those found for other mAbs and suggest that may be why 17T2 is more resilient to spike mutations.

We thank the Reviewer for his/her words of appreciation.

The cryoEM workflow is straight forward. I would recommend moving the methods describing the focused refinement used to obtain a higher resolution structure of the interface from the “Model building and refinement” section to the “Image processing” section.

We agree with the Reviewer and in the **Revised Manuscript** we have moved the Methods section describing the focused refinement used to obtain a higher resolution structure of the interface from the “Model building and refinement” section to the “Image processing” section (page 19).

In looking at the PDB/EMDB validation reports a couple of things are flagged.

1) The reported resolution of 3.46 differs by more than 10% from the resolution estimate based on the FSC curve calculated from the deposited half maps (Validation report p. 11 section 4.2). Could the authors comment on this discrepancy?

In the resolution calculation, multiple FSC curves can be made depending on whether a mask is applied or not. Generally, and particularly for large volumes, the FSC curve that includes masked structures is more accurate. This occurs because calculating the resolution without a mask includes regions that solely consist of noise surrounding the structure. The resolution of 3.46 Å corresponds to the FSC curve with a tight mask (following the criteria outlined in Cryosparc software), which we have selected as the most accurate. In the validation report, the estimated resolution displayed is calculated without a mask. However, it is important to note that the global resolution values, as represented by the FSC curves, are only a general indication of the resolution of the whole reconstruction. The most accurate values are represented in the local resolution map. **Supplementary Figure 4** displays this map, showing that the resolution varies from 3.4 Å to 10 Å.

In response to the Reviewer's valid request, we have included FSC curves under various conditions in the **Revised Manuscript Supplementary Fig 4F** and **4G**. Furthermore, we have made modifications to the corresponding figure caption to provide clearer clarification on this matter.

2) Why in the validation report for EMD-16473, does the FSC never reach 0? When it does in Supplementary Figure 3G in the manuscript?

Our data obtained from Cryosparc shows the following figures: A) curve of final overall and B) curve of locally refined:

A)

B)

The graphic shown in the validation report was generated by the PDB web itself and we are unaware of the protocol followed by PDB staff to generate the FSC curves. We have contacted the PDB staff in order to clarify the figure shown in the validation report. We have received the following answer: “Author-provided FSC is usually from comparison between the two final half-maps that have been masked, whereas the Calculated FSC is based on comparisons of raw unprocessed half-maps. So, we expect there to be a slight difference”.

We have uploaded our FSC data obtained with Cryosparc to the PDB system in order to obtain a new validation report with the author provided FSC, which substitutes the previous one.

- 3) The clash score is high for the associated PDB (25.7). The authors mention that they don't observe side-chain density; however, they don't indicate what kind of restraints were used in the model refinement procedure? If something like Isolde were used with adaptive distance restraints would the clash score improve?

The refinement was performed using Refmac5, applying jelly body restraints, which are indicated for refinement of low-resolution structures. Following the Reviewer's comment, we have used PROSMART software to apply a tighter method and restraint refinement. PROSMART software was employed to create external restraints from homologous structures for the RBD and Fab resolved at high resolution and later applied to the refinement. Unfortunately, this protocol didn't produce a better result and reduced the correlation of the model with the data. Thus, we did not include this new analysis in the **Revised Manuscript**.

Supplementary Table 5 could do with a legend describing some of the acronyms used in the table (d99, CaBLAM etc)

Following the Reviewer's suggestions, we have now included a short description for some of the acronyms used in the “Resolution estimates” and “Model vs. Data” sections (**Supplementary Table 5**). More information on this section can be found in the following article: “Afonine et al, 2018, doi: 10.1107/S2059798318009324”. Furthermore, we would like to note that **Supplementary Table 5** has been replaced with the latest version. It is important to emphasize that this change does not impact the interpretation of the data or the model provided.

What is being reported in the Model vs Data section with the various cross-correlations? How were these calculated?

The various values of model-map correlation coefficients (CC) shown in the Model vs. Data sections are intended to explain the model-map correlation, where as a rule of thumb an arbitrary value of $CC > 0.7$ is considered a good fit and something below 0.5 is considered a poor fit.

Values were calculated using the validation tools included in PHENIX (Adams et al, 2010, doi:10.1107/S0907444909052925). Briefly, and according to the authors' explanations, the aim is to establish a relationship between CC and the coordinate error of an atomic model by placing a model in a P1 box, setting the ADP values to a given value and computing a map (M) of specified resolution from such a model. Then, subjecting this model to a molecular dynamics simulation and CCmask values between M and the maps computed for the models from the simulation trajectory. These CCs are monitored along with the corresponding RMS deviation between the original model and the intermediate model. The MD simulation is continued until the CC reaches zero. This defines the CC as a function of the model deviation. The entire calculation is repeated for different resolutions and ADP values.

What does cc peaks and cc volume refer to?

CCpeaks describes the quality of fit of the strongest peaks in the model and in the map. **CCvolume** uses a mask of points with the highest values in the model map to describe the fit of the molecular envelope defined by the model map.

This information is now contained in the revised **Supplementary Table 5**.

A recently published study in PLOS Biology characterized another neutralizing mAb isolated from convalescent patient infected with the prototype strain (Meng et al., 2023). This study applied similar types of analyses including neutralizing capability, therapeutic efficacy in transgenic mice and structural biology. The discussion section could be expanded to describe how their results compares/contrasts to those presented in this new study.

<https://journals.plos.org/plospathogens/article?id=10.1371/journal.ppat.1011085>

We would like to express our gratitude to the Reviewer for recommending an interesting article that describes another new broadly neutralizing monoclonal antibody (ab08). Unlike the 17T2 mAb, ab08 interacts with the receptor-binding domain (RBD) at a distinct epitope, separate from the receptor-binding motif (RBM). For this reason, we couldn't perform a direct structural comparison between 17T2 and ab08 as we did for S2E12. We have referenced the manuscript when discussing other broadly neutralizing antibodies in the Discussion section of the **Revised Manuscript** (reference 43).

One more final suggestion on how to improve the clarity of the structural figures. Figure 2E, it is hard to see the lines of the H bond network. I would suggest removing the lines and instead leave the residues involved labelled which form the key interactions at the

interface. If you color the sticks according to heteroatom, it will also be easier for the reader to identify the residues you refer to.

We agree with the Reviewer and in the **Revised Manuscript** we have changed **Figure 4E** according to the recommendations. The lines indicating the H bond network have been removed and only the residues involved in the interaction are highlighted. The captions have been corrected according to the changes.

When you describe the increased interface area for the binding site for 17T2, it would be helpful to include quantitative measurements for this, for example buried surface area in PISA.

<https://www.ebi.ac.uk/pdbe/pisa/>

In the original version of the manuscript, we had included information about quantitative measurements of the interaction area between Fab 17T2 and RBD. The passage read as follows: "Fab 17T2/RBD interactions involve both the heavy chain (HC) and light chain (LC) of the antibody, covering 563 Å² for the HC and 295 Å² for the LC of the total interaction surface (**Figure 4D**)" (page 8 in the **Revised Manuscript**). This analysis was performed using the PDBePISA and PDBsum servers. We apologize to the Reviewer for not mentioning the methodology used in the original manuscript. To improve clarity, we have now included a brief description of the methodology used, along with references (references 66-67 in the **Revised Manuscript**) to the servers utilized. The Reviewer can find this addition highlighted on page 19, in the Model Building and Refinement section of the Material and Methods.

REVIEWERS' COMMENTS

Reviewer #1 (Remarks to the Author):

The authors have properly addressed this reviewer's concerns.

Reviewer #2 (Remarks to the Author):

The authors have fully addressed the specific points raised by this referee. Particularly, they have analyzed the neutralization breadth of mAb 17T2 and other antibodies in more detail. 17T2 is able to neutralize more recent variants, although not as potent as earlier variants. The data in the revised manuscript support the conclusion that a new SARS-CoV-2 broadly neutralizing antibody was identified. While the antibody does not seem to be superior over previously identified broadly neutralizing antibodies against SARS-CoV-2, it seems to be an interesting addition to the available collection of SARS-CoV-2 bNAbs.

Reviewer #3 (Remarks to the Author):

all concerns have been addressed in the revised manuscript and I have no further queries.

Response to Reviewer comments

Reviewer #1 (Remarks to the Author):

The authors have properly addressed this reviewer's concerns.

We thank the Reviewer for their positive comment.

Reviewer #2 (Remarks to the Author):

The authors have fully addressed the specific points raised by this referee. Particularly, they have analyzed the neutralization breadth of mAb 17T2 and other antibodies in more detail. 17T2 is able to neutralize more recent variants, although not as potent as earlier variants. The data in the revised manuscript support the conclusion that a new SARS-CoV-2 broadly neutralizing antibody was identified. While the antibody does not seem to be superior over previously identified broadly neutralizing antibodies against SARS-CoV-2, it seems to be an interesting addition to the available collection of SARS-CoV-2 bNAbs.

We thank the Reviewer for their words of appreciation for our work

Reviewer #3 (Remarks to the Author):

All concerns have been addressed in the revised manuscript and I have no further queries.

We thank the Reviewer for their positive feedback.